

**Evolution of Cloud Droplet Temperature and Lifetime in**
**Spatiotemporally Varying Subsaturated Environments with**
**Implications for Ice Nucleation at Cloud Edges**
Puja Roy[1,2], Robert M. Rauber[2], Larry Di Girolamo[2]
[1]Research Applications Laboratory, NSF National Center for Atmospheric Research
[2]Department of Climate, Meteorology & Atmospheric Sciences, University of Illinois Urbana-Champaign
*Correspondence to*:  Puja Roy (pujaroy@ucar.edu)
**Abstract.** Ice formation mechanisms in generating cells near stratiform cloud-tops, where mixing and entrainment
occurs in the presence of supercooled water droplets, remain poorly understood. Supercooled cloud droplet
temperature and lifetime may impact heterogeneous ice nucleation through contact and immersion freezing; however,
modeling studies normally assume droplet temperature to be spatially uniform and equal to the ambient temperature.
Here, we present a first-of-its-kind quantitative investigation of the temperature and lifetime of evaporating droplets,
considering internal thermal gradients within the droplet as well as thermal and vapor density gradients in the
surrounding air. Our approach employs solving the Navier-Stokes and continuity equations, coupled with heat and
vapor transport, using an advanced numerical model. For typical ranges of cloud droplet sizes and environmental
conditions, the droplet internal thermal gradients dissipate quickly ($\leq 0.3$ s) when droplets are introduced to new
subsaturated environments. However, the magnitude of droplet cooling is much greater than estimated from past
studies of droplet evaporation, especially for drier environments. For example, for an environment with pressure of
500 hPa, and ambient temperature far from the droplet of -5°C, the droplet temperature reduction can be as high as
24, 11, and 5°C for initial ambient relative humidities of 10%, 40%, and 70% respectively. Droplet lifetimes are found
to be tens of seconds longer compared to previous estimates due to weaker evaporation rates because of lower droplet
surface temperatures. Using these new end-of-lifetime droplet temperatures, the enhancement in activation of ice-
nucleating particles predicted by current ice nucleation parameterization schemes is discussed.



## 1 Introduction

Ice formation often occurs near cloud tops of stratiform clouds where ice-generating cells are frequently found in a variety of cold, cloudy environments (Plummer et al., 2014; Ramelli et al., 2021). These cells play a crucial role in primary ice nucleation and growth (Tessendorf et al., 2015). Evidence of mixing and entrainment and the presence of supercooled liquid water within and between the highly turbulent cells has been observed (Plummer et al., 2014; Wang et al., 2020; Zaremba et al., 2024). Within regions of entrainment and mixing at cloud boundaries, cloud droplets are exposed to subsaturated environments and undergo evaporation that leads to droplet temperatures that could be several degrees lower than that of the ambient environment (Kinzer and Gunn, 1951; Watts, 1971; Roy et al., 2023). However, in modeling cloud microphysical processes, the difference in temperature between the cloud droplets and their environment is generally assumed to be negligible (Pruppacher and Klett, 1997), i.e., the droplets' temperatures are approximated to be the same as that of their ambient environment. This assumption is reasonable for cloud droplets inside the cloud but breaks down within entrainment and mixing zones at cloud boundaries and may lead to uncertainties in the numerical simulations of microphysical processes. Cloud droplet temperatures affect the calculated droplet diffusional growth or evaporation rates (Roach 1976; Srivastava and Coen 1992; Marquis and Harrington 2005; Roy et al., 2023), and droplet lifetimes (Roy et al., 2023), radiative effects via temperature-dependent refractive indices (Rowe et al. 2020), and ice formation via pathways that require supercooled liquid water droplets, such as contact nucleation (Young, 1974), immersion freezing (Szakáll et al., 2021), and homogeneous nucleation (Khvorostyanov and Sassen, 1998; Khain and Pinsky, 2018). These uncertainties can propagate into microphysical parameterization schemes, leading to possible inadequate representation of mixed-phase cloud properties across various scales (e.g., Large Eddy Simulations (LES), Cloud Resolving Models (CRM), Climate Models), impacting predictions of precipitation or climate change.

Several studies have highlighted the special importance of the air-water interface of the water droplet during ice nucleation. Many experimental and theoretical studies have suggested that ice initiation occurs at the droplet surface (Tabazadeh et al., 2002a; Tabazadeh et al., 2002b; Djikaev et al., 2002; Satoh et al. 2002; Shaw et al., 2005) and the interface thermodynamically favors the contact mode over the immersion freezing mode (Djikaev and Ruckenstein, 2008). Based on their laboratory observations, Tabazadeh et al., (2002a) suggested that homogeneous nucleation of nitric acid dihydrate (NAD) and nitric acid trihydrate (NAT) particles within aqueous nitric acid droplets primarily occurs at the droplet surface. This leads to the hypothesis that phase transformations in atmospheric aerosols may predominantly be surface-based (Tabazadeh et al., 2002b), challenging the traditional theory of homogeneous crystallization where freezing begins inside the volume of the droplet (Volmer, 1939). Satoh et al. (2002) studied cooling and freezing in water droplets due to evaporation in an evacuated chamber and found that droplets rapidly froze with significant supercooling, with the freezing initiated from the droplet surface. Studies employing molecular dynamics simulations (Chushak et al., 1999, 2000) and thermodynamic calculations (Djikaev et al., 2002) additionally corroborate that a crystalline nucleus preferentially forms at the droplet surface rather than within the bulk droplet volume. Laboratory observations from Shaw et al., (2005) reveal that freezing temperatures are 4-5 K higher when an





ice-forming nucleus is closer to the surface of a supercooled water droplet compared to when it's immersed within the
droplet. They found that the nucleation rate at the water surface is significantly higher (by a factor of $10^{10}$) than in the
bulk droplet, indicating that the free energy required for critical ice germ formation decreases when near the air-water
interface, and the jump frequency of molecules from the liquid to the solid phase may be significantly enhanced at the
interface. Lü et al., (2005) conducted ice nucleation experiments with acoustically levitated supercooled water droplets
and found that statistical analyses of nucleation rates indicate ice nucleation predominantly initiates in the vicinity of
the droplet surface. Therefore, given the importance of the droplet surface in ice nucleation and since evaporation is a
surface phenomenon, in the quest to better understand the physical mechanisms responsible for primary ice nucleation,
it is important to accurately investigate the thermal evolution of the evaporating droplet surface as well as the internal
thermal gradients within the supercooled droplet, as ice nucleation is highly temperature dependent.
Few studies in the cloud microphysics literature have carried out explicit numerical estimations and evolutions of
supercooled, evaporating cloud droplet temperatures and lifetimes for a wide range of environmental conditions. Roy
et al., (2023) provides a comprehensive review of past theoretical, numerical, or experimental studies of droplet
evaporation. Most of these studies examined the evaporation of raindrops for above zero temperatures (Kinzer and
Gunn, 1951; Watts 1971; Watts and Farhi, 1975), either assuming steady-state expressions (Beard and Pruppacher,
1971) or simplifying assumptions of linear dependence of saturation vapor density on temperature (Kinzer and Gunn,
1951; Watts 1971; Watts and Farhi, 1975). Srivastava and Coen (1992) assumed the heat storage term in the droplet
heat budget to be negligible, and investigated the evaporation of isolated, stationary hydrometeors by iteratively
solving the steady-state solutions, using saturation vapor pressure relations from Wexler (1976) to calculate the
saturation vapor density. Roy et al., (2023), by including the heat storage term and solving for time-dependent heat
and mass transfer between single, stationary cloud droplets evaporating in infinitely large, prescribed ambient
environments, demonstrated that the temperatures of the cloud droplets (initial radii between 30-50 μm) reach steady-
state quite quickly (within <0.5 s). They considered a wide range of environmental conditions and found that
evaporating droplet temperatures can typically be 1-5 K colder than that of the environment, with values as low as
~10 K for low relative humidity, and low-pressure conditions near 0°C environments. Their steady-state droplet
temperatures agreed well with those of Srivastava and Coen (1992).  They showed that the droplet temperature during
evaporation can be approximated by the thermodynamic wet-bulb temperature of the ambient environment. For most
subsaturated conditions, radiative cooling in cloud-top environments was found to play a negligible role in altering
evaporating droplet temperatures, except for larger droplets in environments close to saturation.
However, two main issues have not yet been accounted for in the aforementioned studies. Firstly, water droplets were
considered to have a uniform bulk droplet temperature, based on the assumption of infinite thermal heat conductivity
of water, thus ignoring the added complexity of simulating the internal thermal gradients within the droplet. (Kinzer
and Gunn, 1951; Watts, 1971; Srivastava and Coen, 1992, Roy et al., 2023). As several studies suggest that the droplet
surface plays a special role in nucleating ice and evaporation being a surface phenomenon, accurate modeling of the
evolution of droplet surface temperature and internal thermal gradients within the droplet volume is required to



correctly predict the ice nucleation rates. Secondly, to date, none of these studies considered the spatiotemporally
evolving effects of thermal and moisture feedback between the droplet and its immediate environment. The rationale
for justifying the usage of constant ambient conditions far away from the droplet was mostly based on studies where
ambient conditions were defined by prescribed temperature and moisture fields far away from a droplet (Sedunov,
1974; Eq. 7.7 of Rogers and Yau, 1989; Srivastava and Coen, 1992). A correction to the ambient conditions at a radius
similar to the mean distance between droplets (~1 mm) was shown to lead to minimal modifications for typical cloud
conditions (Fukuta, 1992). Thus, this assumption holds for droplets distributed homogeneously in space. Concerning
numerically simulating the growth and decay of a droplet population, Grabowski and Yang (2013) stated: "Cloud
droplets grow or evaporate because of the presence of moisture and temperature gradients in their immediate vicinity,
and these gradients are responsible for the molecular transport of moisture and energy between the droplet and its
immediate environment. One may argue that these gradients need to be resolved to represent the growth accurately.
Elementary considerations demonstrate that the moisture and temperature gradients in the droplet vicinity are
established rapidly [i.e., with a characteristic timescale of milliseconds or smaller (e.g., Vaillancourt et al. 2001, and
references therein)]; thus, the steady-state droplet growth equation is accurate enough. More importantly, the volume
affected by these gradients has a radius of approximately 10 to 20 droplet radii.…  One can simply neglect molecular
transport processes in the immediate droplet vicinity and simulate droplet growth using the classical approach, that is,
by applying the supersaturation predicted by the mean (over the volume occupied by the droplet) temperature and
moisture fields…(see Vaillancourt et al. 2001, appendix)."

Here, we quantitatively revisit these arguments within the context of an evaporating supercooled cloud droplet. We
use high-resolution modeling to resolve the spatiotemporally evolving thermal and vapor density gradients in the
vicinity of the droplet as well as include internal heat transfer within the droplet, relaxing the assumption of infinite
thermal heat conductivity of water. Using an advanced numerical model, our framework employs the finite-element
method to solve the Navier-Stokes and continuity equations, coupled with heat and vapor diffusion, with appropriate
boundary conditions. The results from this study extend the findings from Roy et al. (2023) that an evaporating droplet
can exist at a temperature lower than that of the ambient environment, and that the temperature deviation increases
from the steady-state value under certain environmental conditions. This may lead to significant enhancement in ice
nucleation by increasing the predicted number concentrations of activated ice-nucleating particles (INPs) either
immersed within or externally contacting the supercooled droplet. The current study advances the numerical approach
presented in Roy et al. (2023) by including the impact of internal heat gradients within the droplet and spatiotemporally
varying heat and mass transfer between the droplet and its immediate environment. We also provide droplet lifetime
comparisons with estimates from Roy et al. (2023) and pure diffusion-limited evaporation calculations. The
implications of the evaporating supercooled cloud droplet temperatures and lifetimes on ice nucleation at cloud
boundaries are discussed.





**2 Numerical Methodology**

**2.2 Description of COMSOL**

The simulation of the spatiotemporally varying droplet temperature and radius of an evaporating cloud droplet embedded in a gaseous domain is difficult to solve analytically because of the moving and shrinking boundary at the surface of the evaporating droplet. These kinds of moving boundary problems are also known as Stefan problems. To model this process, we have used an advanced numerical solver, COMSOL (Version 6.0), which employs a finite element method to solve partial differential equations (PDEs). The Navier-Stokes and Fick's second law of diffusion equation, which follows from the continuity equation, along with appropriate boundary conditions (see Sec. 3) are solved to conserve mass and momentum in the whole system. The PDEs are discretized and solved along non-uniform moving mesh nodes using the Arbitrary Lagrangian-Eulerian technique (Yang et al., 2014) to accurately track the moving air-water interface at the droplet surface.

The COMSOL multiphysics software uses cylindrical coordinates ($r$, $\phi$, $z$) to solve 2D axisymmetric geometries ($z$-axis is the axis of symmetry), where $r$ represents the radial distance from the longitudinal axis, $\phi$ is the azimuthal angle (in the interval from $-\pi$ to $\pi$), and $z$ is the distance from the origin along the longitudinal axis (COMSOL 2023a). For this modeling scenario, the geometry consists of a 2D axisymmetric domain with the center of the cloud droplet at the origin (defined at $r = 0$, $z = 0$) with ambient air surrounding the droplet (Fig. 1). The physics interfaces take care of the differential operators while solving the equations arising from the conservation laws. The following physics interfaces in COMSOL were used to simulate droplet evaporation: (1) *Two-Phase Laminar Fluid Flow*, which includes a moving mesh to track the shrinking water–air interface of the evaporating water droplet and fluid–fluid interface that incorporates evaporative mass flux; (2) *Transport of Diluted Species* to track water vapor diffusion through the air domain and predict the evaporation rate at the droplet surface; and (3) *Heat Transfer in Fluids* which accounts for the non-isothermal flow within the computational domain, temperature-dependent saturation vapor density at the droplet interface, and a boundary heat source to account for the latent heat of evaporation. The computational domain also includes an infinite element air domain (COMSOL 2023b) to specify and maintain boundary conditions far away from the droplet. The physics modules are coupled through non-isothermal flow between heat transfer and fluid flow, and mass transport at the fluid–fluid interface between fluid flow and species transport.

The non-uniform moving mesh, created by breaking down the computational domain into numerous fine elements of variable sizes, uses the Arbitrary Lagrangian-Eulerian technique (Yang et al., 2014) to accurately track the moving air-water interface at the droplet surface. In this study, we have used triangular mesh elements (COMSOL 2023c) within the droplet and quadrilateral mesh elements (COMSOL 2023d) for the rest of the domain as shown in Fig. 1. Finally, to simulate the water droplet evaporating in ambient air system, with appropriate initial and boundary conditions, the discretized PDEs are then numerically solved along each of the mesh nodes with adaptive time steps





(≤0.01s) to maintain numerical stability and obtain the solution (the temporal evolution of droplet temperature and
radius) for a range of conditions.

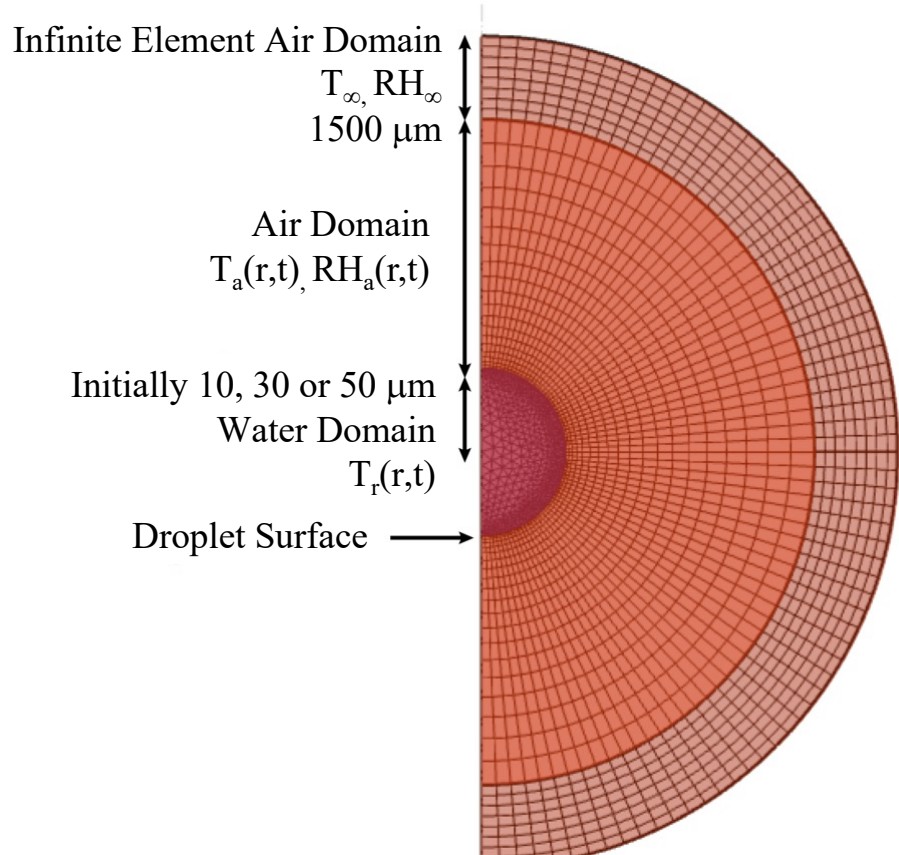

**Figure 1: Computational domain (not to scale) including the evaporating droplet, embedded in the air domain.**

**2.2 Justification for choice of environmental parameters in the simulations**

Probing the evolution of the droplet and its immediate environment under a wide swath of conditions was
computationally too expensive, thus, certain choices regarding the parameter selection were made. The assumption
behind the computational set-up is that the supercooled droplet is suddenly introduced to a subsaturated environment
with ambient temperature, $T_\infty$ = 273.15 K, 268.15 K, or 263.15 K, as might happen when the droplets are near cloud
boundaries such as those occurring in cloud-top generating cells. These temperatures are the ones where activation of
INPs is thought to be least effective. Calculations presented in Sec. 4 consider three different environments having





ambient relative humidity, $RH_\infty$ = 10, 40, and 70%, and two different ambient pressures, $P$ = 500, and 850 hPa, and
initial cloud droplet radii, $r_0$, of 10-50 μm. The pressure levels were chosen based on the occurrence of 273.15 K,
268.15 K, and 263.15 K in standard atmospheric profiles for tropical latitudes and middle latitudes under warm and
cool season conditions (Standard Atmosphere, 2021). Overall, 90 numerical experiments were performed using
various combinations of initial $RH_\infty$, $T_\infty$, $P$, and $r_0$ to obtain a better understanding of the relationships between the
evolution of droplet temperatures and radii, and environmental variables. Of these, the results of 54 experiments are
reported in detail herein. The results of these experiments are later summarized in Figs. 4-14 and Tables 1-2. The
specific combinations of environmental parameters and initial droplet radii used in this study were also selected to
enable easy comparison with results from a previous study of droplet evaporation (Roy et al., 2023). Also, to be noted,
the effect of radiation in this study was neglected based on the Roy et al. (2023), which demonstrated the negligible
role played by radiation in modifying evaporating droplet temperatures under most subsaturated conditions (RH <

200    80%).


**202    2.3 Justification for choice of droplet lifetime cut-off**


For each experiment, the computational time rose exponentially to maintain numerical stability as the droplet radius
decreased during evaporation and the grid sizes needed to be smaller. To avoid exceptionally long computation time,
the cut-off radius for the simulations was set to be when the volume of the droplets decreased by 99.5% to reach 0.5%
of the initial droplet volume. For $r_0$ = 10, 20, 30, 40, 50 μm, the cutoff radii of the droplets are 1.71, 3.42, 5.13, 6.84,
and 8.55 μm, respectively. Note that due to the Raoult effect, for a solution droplet with a mass of dissolved and
ionized NaCl = $10^{-13}$ g, the reduction in the evaporation rate ($dr/dt$) from that of a pure water droplet is about 1% for
a 1 μm radius droplet and 4% for a 0.7 μm droplet. As all cut-off radii considered here are > 1 μm, the solute effect
can be neglected. From the Kelvin equation, the equilibrium vapor pressure over a curved surface of pure water
approaches the value of equilibrium vapor pressure over a flat surface of pure water for a radius > 0.01 μm. Thus,
curvature effects were also neglected. For simplicity, we will refer to the cutoff time as the *droplet lifetime*, although
the droplets will survive for a longer time before complete evaporation. The droplet lifetime increases with the initial
droplet radius, higher atmospheric pressure, and higher $RH_\infty$ (Fig. 2).





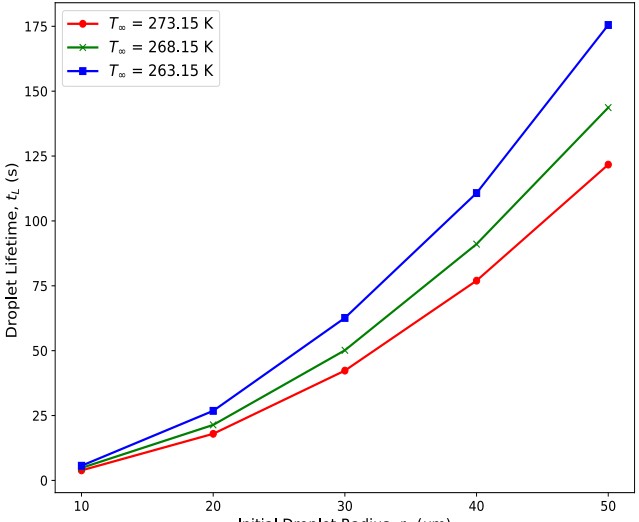


**Figure 2: Droplet lifetimes, $t_L$ in seconds, for droplets with varying initial droplet radii, $r_0$ = 10, 20, 30, 40 and 50 μm,**
**evaporating in an initial ambient environment with three different ambient temperatures, $T_\infty$ = 273.15 (0°C), 268.15 (-5°C)**
**and 263.15 (-10°C) K, with relative humidity, $RH_\infty$ = 70%, with pressure, $P$ = 850 hPa.**

**2.4 Sensitivity to domain size**

It was important to ensure that the spatiotemporally varying thermal and vapor density gradients in the ambient air in
the vicinity of the evaporating droplet don't interfere with the constant ambient conditions ($RH_\infty$ and $T_\infty$) at the external
boundary of the computational domain. Sensitivity tests with different air domain sizes of 10, 30, and 50 times the
initial droplet radius were carried out to determine the droplet temperature and radial dependence on domain size. It
was found that the evolution of droplet temperature and radius was not sensitive to domain sizes larger than 10 times
the droplet radius considered. Based on the sensitivity analysis, the maximum size of the computational domain for
all experiments was fixed at 1500 μm, 30 times the largest droplet considered.

**3 Theory**
**3.1 Assumptions**

The framework of the numerical model assumes that an isolated, stationary, spherical, pure water droplet is suspended
within a 2D axisymmetric ambient air domain with constant ambient temperature (≤ 0°C) and relative humidity
(<100%) at a sufficiently far distance away from the droplet that the droplet evaporation does not influence the far
environment. The water droplet and air are considered to be Newtonian fluids, with the assumption that no internal
circulation occurs within the droplet and that there is no ventilation, no radiative heat transfer, and negligible buoyancy
effects due to gravity. This computational approach is an advanced form of the one described in Roy et al., (2023),



but also includes the effect of internal droplet heat transfer and spatiotemporal gradients in temperature and vapor
density between the droplet and the environment (see discussion in Sec. 5).

**3.2 Governing Equations**

Based on the above assumptions, the following are the equations governing the system during droplet evaporation in
the ambient air.

(1) Fluid flow: The *Laminar Flow* interface models the weakly compressible form of the Navier-Stokes equation,
along with the continuity equation in the water and air domains,

$$\rho \frac{\partial \boldsymbol{u}}{\partial t} + \rho(\boldsymbol{u}\cdot\nabla)\boldsymbol{u} = \nabla \cdot[-\mathrm{p}\boldsymbol{I} + \boldsymbol{\tau}] + \boldsymbol{F}$$

$$\boldsymbol{\tau} = \mu(\nabla\boldsymbol{u} + (\nabla\boldsymbol{u})^{\boldsymbol{T}}) - \frac{2}{3}\mu(\nabla\cdot\boldsymbol{u})\mathbf{I}$$

$$\frac{\partial \rho}{\partial t} + \nabla\cdot(\rho\boldsymbol{u}) = 0$$


where $t$ is time, $\rho$ is the fluid density (kg/m³), $\boldsymbol{u}$ is the fluid velocity vector (m/s), $p$ is pressure (Pa), $\boldsymbol{I}$ is the identity
tensor, $\boldsymbol{\tau}$ is the viscous stress tensor (Pa), $\mu$ is the fluid dynamic viscosity, $\boldsymbol{F}$ is the external volume force vector (N/m³),
which is assumed to be negligible here.


(2) Heat Transport: The *Heat Transfer in Fluids* interface models heat transfer in all domains (air, water, infinite
element domain) using the following version of the heat equation:

$$\rho C_p \frac{\partial T}{\partial t} + \rho C_p \boldsymbol{u}\cdot\nabla T + \nabla\cdot\boldsymbol{q} = Q_b$$

$$\boldsymbol{q} = -k\nabla T$$


where $\rho$ (kg/m³) is the fluid density, $C_p$ (J/(kg·K)) is the fluid heat capacity at constant pressure, $T$ is the temperature,
$k$ (W/(m·K)) is the fluid thermal conductivity, $\boldsymbol{u}$ (m/s) is the fluid velocity field from the Laminar Flow interface, $\boldsymbol{q}$
(W/m²) is the heat flux by conduction, and $Q_b$ (W/m³) is the heat sink due to evaporative cooling at the droplet surface.

(3) Mass transport: The *Transport of Diluted Species* interface models water vapor transport through Fick's laws of
diffusion, solving the mass conservation equation for vapor transfer in all domains except within the cloud droplet:
$$\frac{\partial c}{\partial t} + \nabla\cdot\boldsymbol{J} = 0$$

$$\boldsymbol{J} = -D\nabla c$$




where $c$ is the concentration of water vapor (mol/m$^3$), $D$ denotes the diffusion coefficient (m$^2$/s), and $J$ is the mass flux
diffusive flux vector (mol/(m$^2$·s)). $D$ is calculated following Hall and Pruppacher (1976) and defined as follows: $D =$
$0.0000211 \frac{P_0}{P} \left[\frac{T}{T_0}\right]^{1.94}$ (m$^2$ s$^{-1}$) with reference pressure, $P_0 = 1013.25$ hPa, reference temperature, $T_0 = 273.15\ K$,
atmospheric temperature, $T$, and pressure, $P$. In this study, values of $P$ are either fixed at 500 or 850 hPa to determine
the effect of ambient air pressure on droplet evaporation. $J$ is obtained from the Laminar Flow interface through
coupling between these interfaces.

**3.3 Initial conditions**

The initial velocity components in the r, and z directions are assumed to be 0 m/s in both air and water domains. The
initial fluid pressure is $p = P_{0,air}$ (Pa), specified either at 500 or 850 hPa in the air domain, and in the water domain, $p$
$= P_{0,water} = \frac{2\sigma}{r_0}$ Pa, where surface tension, $\sigma = 70 \times 10^{-3}$ (N/m). For the heat transfer module, all domains are assumed
to be at a prescribed initial ambient temperature, $T_0$, which is the same as that of a point at a far distance away from
the droplet, $T_\infty$ . For the vapor transfer interface, except within the droplet, all domains are at an initial vapor
concentration of $c_{0,air}$ which is again assumed to be the same as that of the constant ambient concentration value far
from the droplet, $c_\infty$, calculated as follows:
$c_\infty = \frac{RH_\infty \times e_{S T_\infty}}{R_{univ} \times T_\infty}$ where, $RH_\infty$ is set at a constant ambient relative humidity far from the droplet, $R_{univ} = 8.3145$ (J/mol/K)
and saturation vapor pressure, $e_{S T_\infty} = 610.94 * \exp\left(\frac{17.625*T_\infty}{T_\infty+243.04}\right)$ (in Pa, with $T_\infty$ in °C) following Alduchov and
Eskridge (1996).


**3.4 Boundary Conditions**

1. At the center of the domain, $r = 0$, axisymmetric conditions are applicable:

$$\boldsymbol{u} \cdot \boldsymbol{n} = 0$$
$$[-p\boldsymbol{I} + \boldsymbol{\tau}] \cdot \boldsymbol{n} = 0$$
$$\boldsymbol{q} \cdot \boldsymbol{n} = -k\nabla T \cdot \boldsymbol{n} = 0$$
$$-D\nabla c \cdot \boldsymbol{n} = 0$$
where $\boldsymbol{n}$ is the outward-pointing surface normal vector.

2. At the fluid-fluid interface i.e., droplet-air boundary, the droplet surface is assumed to be at vapor saturation
throughout its lifetime. Hence, saturated vapor concentration at the shrinking droplet boundary, using the ideal gas





law, is given by, $c_{sat}(T_{sf}) = \frac{e_s(T)}{R_{univ} \times T}$ where saturation vapor pressure, $e_s(T)$, is estimated following Alduchov and
Eskridge (1996) at $T = T_{sf}$, the temperature at the droplet surface (in °C).

The local evaporative mass flux at the interface is given by diffusion of water vapor across the water-air interface, $M_J$
(kg/ m² s)
$$M_J = M_w \boldsymbol{n} \cdot (-D\nabla c)$$


where the molecular weight of water, $M_w = 0.018$ (kg/mol). Although the temperature is continuous across the droplet-
air boundary, there is a discontinuity in heat flux across the interface due to the evaporation of water. Thus, the latent
heat of evaporation $L$, defined as $L = [2501 - 2.44T_r]$ kJ kg⁻¹ with droplet surface temperature, $T_r$ in °C, is
incorporated as a boundary heat sink as $-M_J L$ (W/m²).

The mass balance at the water-vapor boundary at the droplet surface, and the velocity of the moving mesh $\boldsymbol{u_{mesh}}$, at
the shrinking water-air interface, are expressed by the following equations, based on Scardovelli and Zaleski, (1999):

$$\boldsymbol{u_w} = \boldsymbol{u_v} + M_J\left(\frac{1}{\rho_w} - \frac{1}{\rho_v}\right)\boldsymbol{n}$$


$$\boldsymbol{u_{mesh}} = \left(\boldsymbol{u_w} \cdot \boldsymbol{n} - \frac{M_J}{\rho_w}\right)\boldsymbol{n}$$


where the subscripts $w$ and $v$ represent water and vapor respectively.

The stresses are balanced at the water-vapor interface by the following conditions:

$$\boldsymbol{n} \cdot (\mathbf{S_w} - \mathbf{S_v}) = \sigma(\nabla_\sigma \cdot \boldsymbol{n})\boldsymbol{n} - \nabla_\sigma \sigma$$

$$\mathbf{S} = [-\mathrm{p}\boldsymbol{I} + \boldsymbol{\tau}]$$


where $\mathbf{S}$ is the total stress tensor and $\nabla_\sigma$ is the surface gradient operator defined by

$$\nabla_\sigma = (\boldsymbol{I} - \boldsymbol{n} \cdot \boldsymbol{n}^T)\nabla$$


In the normal direction of the boundary, the force is balanced by,

$$\boldsymbol{n} \cdot (\mathbf{S_w} - \mathbf{S_v}) = \frac{\sigma}{r_c} \cdot \boldsymbol{n}$$




where $r_c$ is the curvature radius.

3. The external air domain boundary is open with the following condition:
$[-\mathrm{p}\mathbf{I} + \boldsymbol{\tau}]\boldsymbol{n} = -f_0\boldsymbol{n}$, where normal stress, $f_0 = 0$ N/m².

4. The infinite element domain consists of air and is considered to be an ideal gas. The temperature, relative humidity,
and concentration far from the droplet i.e., at the inner boundary of the infinite element domain, are fixed at $T_\infty$ and
$c_\infty$, respectively.

**3.5 Coupling between the COMSOL interfaces**

To numerically model the evaporating droplet embedded in the air domain, intercoupling between the three physics
interfaces - laminar two-phase flow (formulated within the Arbitrary Lagrangian-Eulerian framework), the heat
transfer in fluids, and the transport of diluted species within the air medium are established through the following
mechanisms: (i) the local evaporative mass flux at the droplet-air interface, which is related to the mesh velocity for
the laminar flow, is estimated by the diffusion of water vapor in the air domain; (ii) saturated vapor concentration at
the droplet-air interface, which serves as a boundary condition for the vapor diffusion, is calculated using the local
temperature at the droplet interface; and (iii) the evaporative heat flux at the droplet-air interface acts as a heat sink
boundary condition for the heat transfer in fluids module.

**4 Results**

**4.1 Internal Droplet Temperature Evolution**

Since evaporation is a surface phenomenon, with the evaporative cooling at the droplet surface acting as a heat sink,
the temperature of the evaporating droplet surface should be lower than the center of the droplet. This is indeed the
case, as shown in the examples in Fig. 3, where 10, 30 and 50 μm droplets are evaporating in two types of
environments: very dry ($RH_\infty = 10\%$) and relatively moist ($RH_\infty = 70\%$), with $P = 500$ hPa, and $T_\infty = 273.15$ K. Note
that the center to surface temperature gradient within the droplet forms almost instantaneously (< smallest output
timestep of 0.01 s) as evaporative cooling at the droplet surface occurs extremely fast. The time required for the droplet
to reach internal thermal equilibrium depends slightly on the initial size of the droplet and the ambient $RH_\infty$, with
larger droplets and drier environments leading to more time required by the droplets to reach equilibrium. However,
generally, for typical cloud droplet sizes and environmental conditions considered here ($r_0 = 10, 30, 50$ μm), the
internal thermal gradients dissipate and the temperatures throughout the droplets become uniform in $\leq 0.3$ s. This can
be explained by the high thermal conductivity values of water (assumed constant at 0.556 W/(m K)) and the absence
of any heat source within the droplet. For this study, we have simulated internal droplet heat transfer for the entirety



of the droplet lifetime and will be reporting the average droplet temperatures as "droplet temperatures" in the results,
unless noted otherwise.

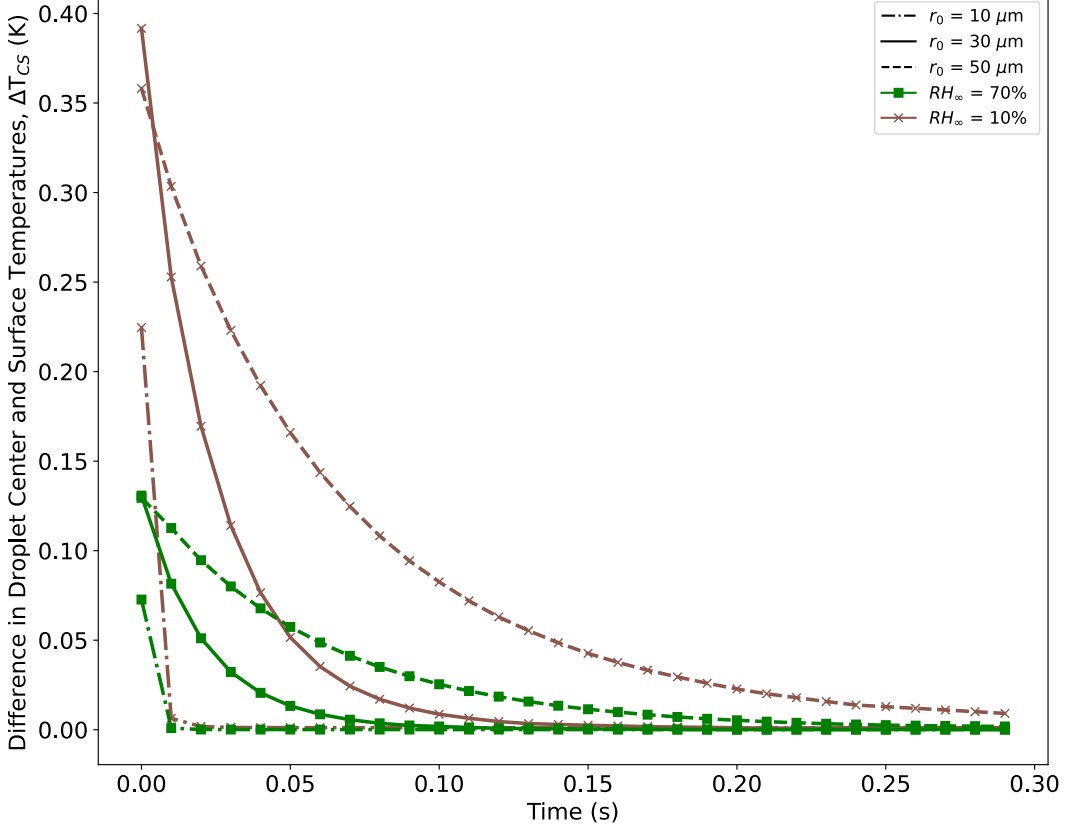

**Figure 3: Thermal evolution of the temperature difference between the droplet center and surface temperatures, $\Delta T_{CS}$**
**(K), for $r_0$ = 10, 30 and 50 μm, for two kinds of environments – dry ($RH_\infty$ = 10%, brown curves) and relatively moist ($RH_\infty$**
**= 70%, green curves), with $P$ = 500 hPa, $T_\infty$ = 273.15 K (0°C).**
**4.2 Droplet Thermal and Radial Evolution: Influence of Initial Droplet Size and Environmental Factors**

Figures. 4 and 5 depict the evolution of the droplet average temperatures and radii ($r_0$ = 10, 30 and 50 μm) for the first
10 seconds of their lifetimes (as defined in Sec. 2c), for different environments with constant ambient conditions ($T_\infty$,
$RH_\infty$, and $P$) far from the droplet. These figures also visually summarize droplet temperatures at the end of their
lifetimes ($T_L$) and the total lifetimes of the droplets ($t_L$). For all numerical experiments, the evaporating droplet
temperature decreases sharply, within < 0.5 s, to a certain temperature defined here as the inflection point in the curves,
$T_i$ (see discussion in Sec. 4c and Sec. 5a). After reaching $T_i$, the decrease in droplet temperature is relatively more
gradual as can be seen from Figs. 4 and 5. For example, in Fig. 4(c), for $P$ = 500 hPa, $T_\infty$ = 268.15 K (-5°C), $RH_\infty$ =
10%, a droplet with $r_0$ = 10 μm, takes about 0.03 s to reach $T_i$, at 260.98 K (a decrease of 7.17 K from initial



temperature, with a mean cooling rate of 239 K s$^{-1}$). In contrast, a 30 µm droplet takes about 0.12 s to reach $T_i$, at
260.85 K (a decrease of 7.3 K from initial temperature, with a mean cooling rate of 60.83 K s$^{-1}$), and a 50 µm droplet
takes about 0.33 s to reach $T_i$ (with a mean cooling rate of 22.12 K s$^{-1}$). Finally, the 10 µm droplet reaches the end of
its lifetime in 1.05 s i.e. $t_L$ = 1.05s with temperature, $T_L$ = 244.12 K, with a mean cooling rate of 16.52 K/s after
reaching $T_i$, while for the 30 µm droplet, $t_L$ = 11.4 s with $T_L$ = 244.31 K (mean cooling rate of 1.47 K/s after reaching
$T_i$), and $t_L$ = 32.76 s for the 50 µm droplet with $T_L$ = 244.29 K, with a mean cooling rate of 0.51 K/s after reaching $T_i$.

In general, we can see that a higher ambient $T_\infty$, and lower $RH_\infty$ and $P$ leads to a larger reduction in droplet temperature
from its initial temperature. Therefore, drier, relatively warmer (closer to 0°C), and lower-pressure environments lead
to the strongest evaporative cooling of the droplets. Also, due to evaporative cooling, the droplets survive longer as
compared to the pure diffusion-limited evaporation approach where the decreases in evaporating droplet temperature
have not been considered (see Sec. 5). However, drier, relatively warmer (close to 0°C), and lower-pressure
environments lead to smaller droplet lifetimes as compared to more humid environments, with lower ambient
temperatures and higher pressures.

**4.3 Environmental Evolution: Evolution of Temperature, Relative Humidity, and Wet-Bulb Temperature in**
**the Air domain near the droplet**

Figures. 6-8 (a, d) show radial cross sections of the computational domain, starting from the center of the droplet on
the origin of x axis = 0 µm to the edge of the domain at x = 1500 µm, while Figs. 6-8 (b, e) expand the dashed box
regions of Figs. 6-8 (a, d), and Figs. 6-8 (c, f) further expand the dashed box regions of Figs. 6-8 (b, e). All panels
show the spatiotemporal evolution of temperature, relative humidity, thermodynamic wet-bulb temperature, and
droplet radius for a droplet with initial radius, $r_0$ = 50 µm, introduced to an initial environment with pressure, $P$ = 500
hPa, ambient temperature, $T_\infty$ = 268.15 K (-5°C), with two different relative humidities, $RH_\infty$ = 10% and 70%. The
evolution of temperature within the droplet is left of the dashed black line, which denotes the droplet radius.

As the droplet evaporates in the subsaturated domain, evaporative cooling occurs at the droplet surface, leading to
heat transfer both from within the warmer droplet and the surrounding air to balance the cooling at the droplet surface.
Since the droplet has no constant internal heat source, the internal thermal gradients dissipate quite fast (within 0.3 s)
and the average droplet temperatures continue to decrease as the droplet evaporates. Due to heat exchange between
the droplet surface and the ambient air in its vicinity, transient thermal gradients in the ambient air develop and lead
to a decrease in the air temperature near the droplet. As the droplet shrinks in size along with cooling further, the
colder envelope of air surrounding the droplet shrinks as well and the ambient air far from the droplet, at a constant
temperature, acts as a heat source and supplies heat to the rest of the domain to equilibrate the air temperature.
Comparing Fig. 6 (a) and (d), at the lower $RH_\infty$, the magnitude of evaporative cooling is much higher. For example,
the average temperature of the 50 µm droplet decreases by ~ 10 K in 9 s when $RH_\infty$ = 10%, while the decrease is ~ 5
K in 120 s, when $RH_\infty$ = 70%.



**Figure 4: Droplet temperature evolution (left column) and radius evolution (right column) for three different $RH_\infty$ ($RH_\infty$ = 10% (brown curves), 40% (orange curves) and 70% (green curves)), three different $r_0$ ($r_0$ = 10 μm (dot-dashed lines), 30 μm (solid lines) and 50 μm (dashed lines)), with three different $T_\infty$ = 273.15 K (0°C) (a, b), 268.15 K (-5°C) (c, d) and 263.15 K (-10°C) (e, f), for $P$ = 500 hPa. For each $RH_\infty$, the droplet temperature at the end of its lifetime ($T_L$, in K) is given in (a,c,e) and the time taken to reach the end of its lifetime ($t_L$, in s) is given in (b,d,f).**

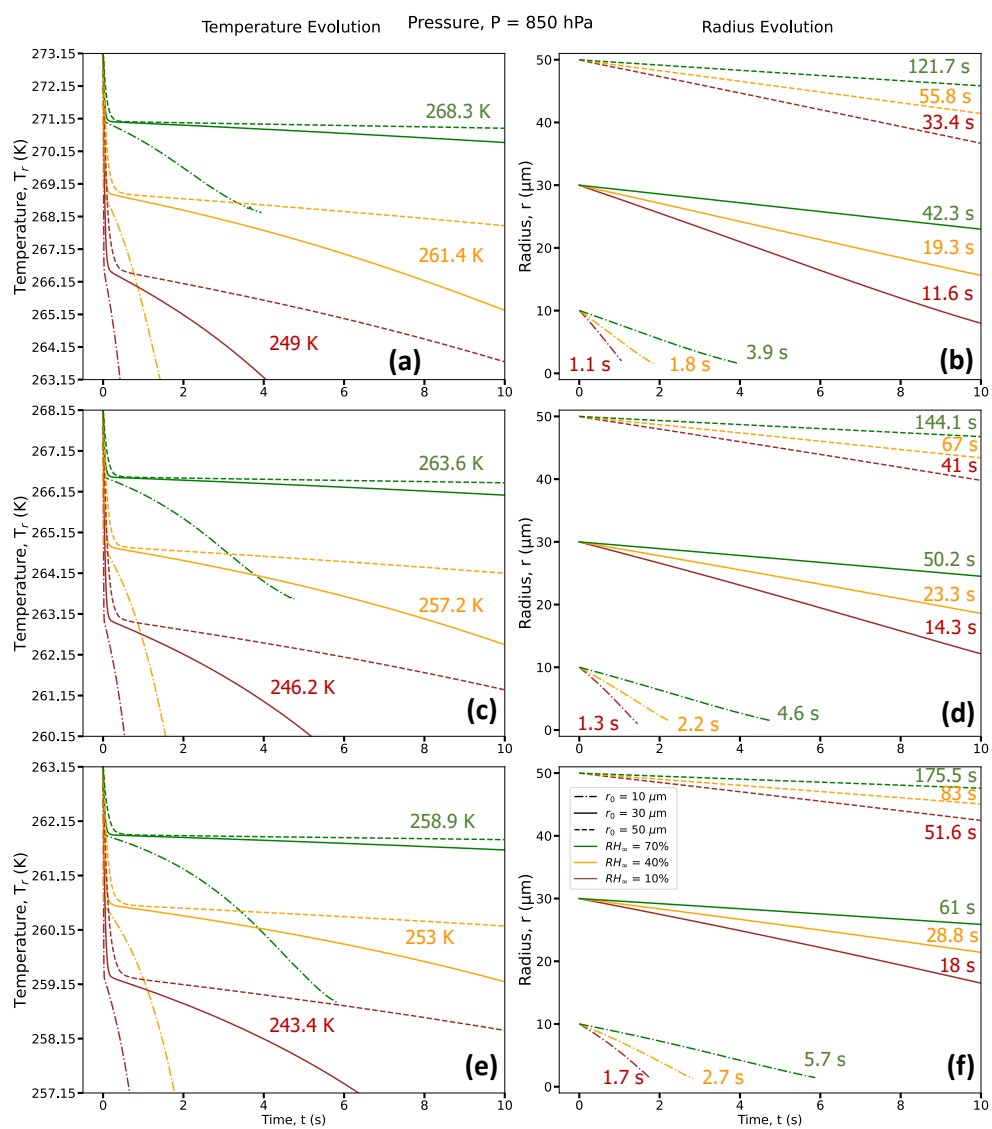

**Figure 5: Same as Fig. 4 but for *P* = 850 hPa.**

In these simulations, the air in contact with the droplet surface is saturated with respect to water, i.e., *RH* = 100% (Fig. 7, a-f), consistent with assumptions of isolated, stationary evaporating droplets (Kinzer and Gunn, 1951; Srivastava and Coen, 1992). As the water vapor from the evaporating droplet surface diffuses into the surrounding environment, with an initial *RH* (same as *RH∞*) of say 10%, vapor density gradients, similar to the thermal gradients, appear and impact the immediate environment of the droplet. These spatiotemporally varying thermal and vapor density gradients play an important role in affecting the droplet temperatures, evaporation rates, and in turn, droplet lifetimes.

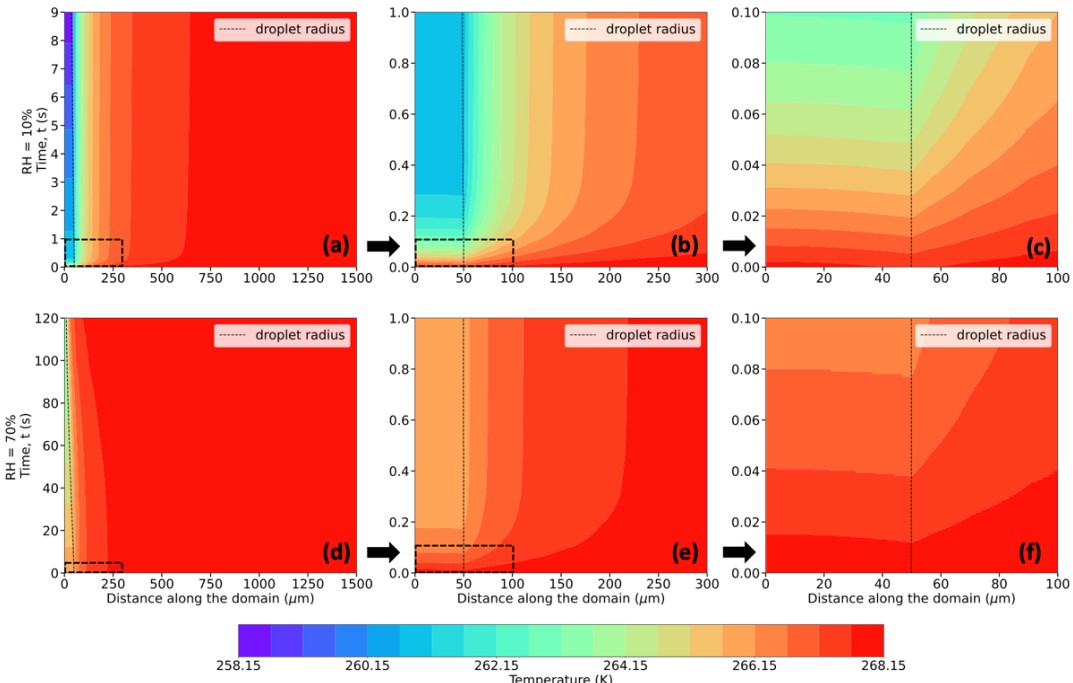

**Figure 6: Evolution of temperature (in K, shaded contours), and droplet radius (in μm, dashed black trace) for a 50 μm**
**droplet, immersed in an environment with $T_\infty$ = 268.15 K (-5°C), $P$ = 500 hPa, and $RH_\infty$ = 10% (top row) and 70% (bottom**
**rows). Figures denoted as (b) and (e), and (c) and (f) present zoomed-in plot areas marked by the dashed boxes in (a) and**
**(d), and (b) and (e), respectively.**




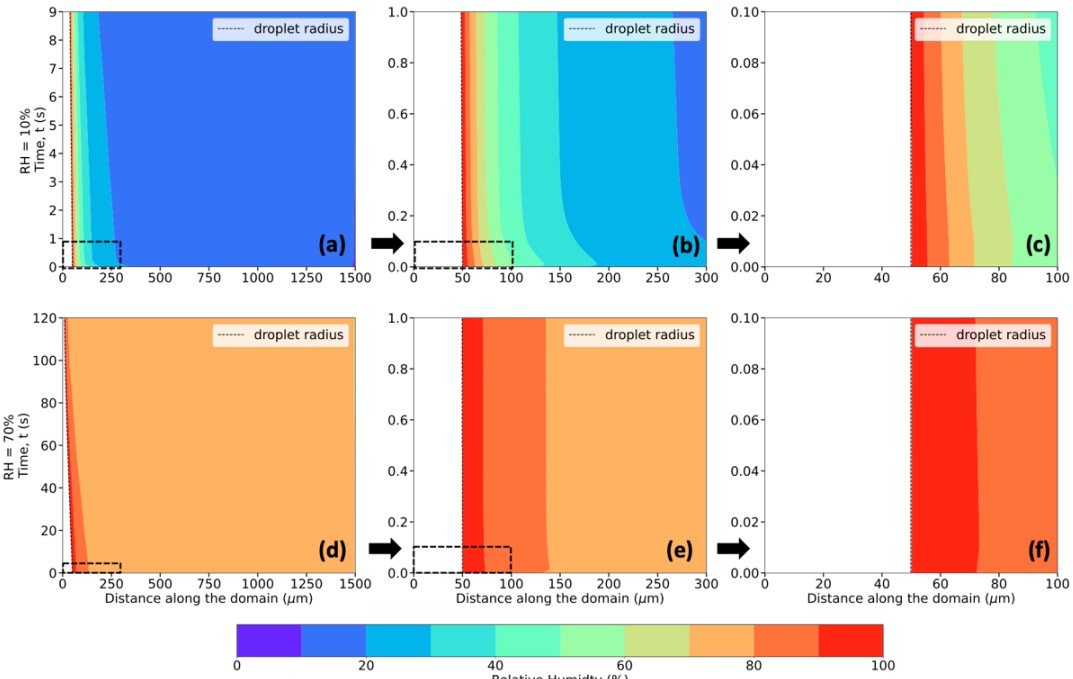


**Figure 7: Same as Figure 6, but for Relative Humidity (in %, shaded contours), instead of Temperature.**

Roy et al. (2023) has shown that an evaporating cloud droplet temperature can be well-approximated by the
thermodynamic wet-bulb temperature of the environment, especially at higher relative humidities and pressures, and
lower ambient temperatures. Following the iterative procedure used in Roy et al. (2023) to calculate the
thermodynamic wet-bulb temperature ($T_{WB}$), Fig. 8 (a-f) depicts the evolution of $T_{WB}$ of the surrounding environment.
Unlike previous studies (Srivastava and Coen, 1992; Roy et al., 2023), the ambient environment in this study is not
assumed to be spatiotemporally invariant. Hence, as the thermal and vapor density gradients evolve in the ambient air,
the $T_{WB}$ of the environment evolves as well, depending on the temperature, relative humidity, and pressure, with the
droplet surface temperature the same as that of the $T_{WB}$ of its immediate environment at all times. Of interest, the
droplet temperature decreases very quickly to $T_i$ within < 0.5 s (Figs. 4 and 5), which agrees very well with the initial
$T_{WB}$ of the surrounding environment and the constant value of the thermodynamic wet bulb temperature far from the
droplet ($T_{WB\infty}$). For example, in Fig. 8(a-c), $T_\infty$ = 268.15 K, $P$ = 500 hPa, $RH_\infty$ = 10%, $T_{WB\infty}$ = 261.64 K, and in Fig.
8(d-f), for $RH_\infty$ = 70%, $T_{WB\infty}$ = 266.13 K. Fig. 8 shows the two phases of the evolution of $T_{WB}$ of the immediate
environment for two $RH_\infty$ environments – initially, there is a very fast decrease of the air temperature at the droplet
surface to $T_{WB\infty}$ typically within < 0.3 s, and then a more gradual decrease of $T_{WB}$ at the droplet surface as the thermal
and vapor density gradients in the ambient air become relatively steadier and more established for a period of time,
and as their spheres of influence start shrinking as the droplet starts getting smaller in size.



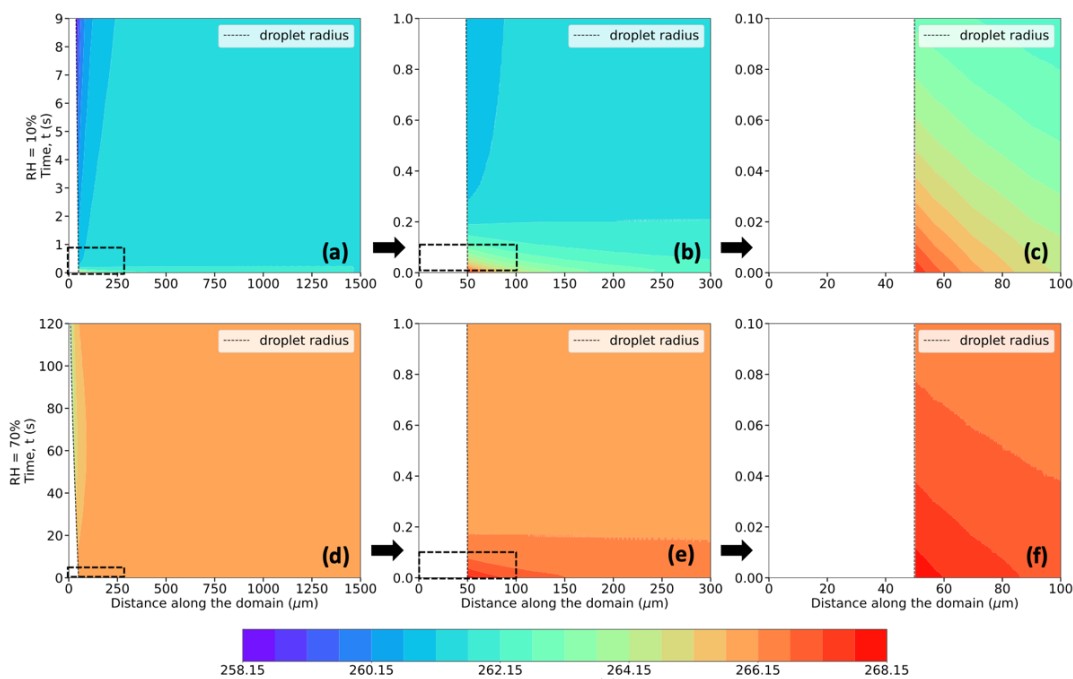


**Figure 8: Same as Figure 6, but for thermodynamic wet-bulb temperature (in K, shaded contours).**

**4.4 Influence of initial droplet size and ambient environmental factors on the thermal evolution of the droplet**
**and its surrounding environment**

To capture the overall trends spanning most of the parameter space, Figs. 9-14 and Tables 1-2 summarize the results
from 54 numerical experiments, using various combinations of ambient conditions ($RH_\infty$, $T_\infty$, and pressure, $P$, and $r_0$)
specified at a distance far away from the droplet.

**4.4.1 Effect of Ambient Relative Humidity, $RH_\infty$**

The decrease in droplet temperature is larger when the $RH_\infty$ is lower due to higher evaporation rates and stronger
evaporative cooling under drier conditions. For instance, as shown in Table 1 and Fig. 9 (a, b, c), 30 μm droplets reach
~ 247.3 K (a decrease of 25.8 K from the initial temperature of 273.15 K) for $RH_\infty$ = 10%, ~ 261.1 K (a decrease of
12.1 K) for $RH_\infty$ = 40% and ~ 268.2 K (a decrease of ~ 5 K) for $RH_\infty$ = 70%. The droplet lifetimes vary widely
depending on $RH_\infty$, with lifetimes increasing with an increase in humidity. For example, the droplet lifetimes for the
30 μm droplet are ~ 9.5 s, 16.7 s, and 37.3 s for environments with $RH_\infty$ = 10%, 40% and 70%, respectively (Table
2). The decrease in droplet temperature and increase in droplet lifetime show similar dependence with increasing $RH_\infty$
for 10 and 50 μm droplets as well.



**4.4.2 Effect of Initial Droplet Size, $r_0$**

From Figs. 9-14, the decrease in droplet temperatures is independent of the initial droplet size if all other initial
environmental conditions are kept constant. For example, from Table 1 and Fig. 10 (a-i) at $P$ = 500 hPa, 10, 30 and
50 μm droplets reach ~ 244 K (a decrease of ~ 24 K from the initial temperature of 268.15 K) for $RH_\infty$ = 10%, ~ 256.8
K for $RH_\infty$ = 40%, and ~ 263.5 K for $RH_\infty$ = 70%. On the other hand, the droplet lifetime strongly depends on the
initial droplet size, as the larger droplets take more time to evaporate as compared to the smaller ones. For
environments with $RH_\infty$ = 10%, 40% and 70%, the droplet lifetimes for the 10 μm droplet are ~ 1.1 s, 1.8 s, and 3.9 s,
while for the 30 μm droplet are ~ 11.4 s, 19.4 s, and 42.8 s, and for the 50 μm droplet are ~ 32.8 s, 55.8 s, and 123.1
s, respectively (Table 2). For a higher pressure of $P$ = 850 hPa (Table 1 and Fig. 13a-i), at the same $T_\infty$, irrespective
of $r_0$, the decrease in droplet temperatures is slightly smaller as compared to $P$ = 500 hPa, with values of 22 K, 11 K,
and 4.6 K. The nature of these dependencies on $r_0$ is in good agreement with those reported in Roy et al., (2023). The
radial dependence of the thermal gradients in the ambient air also depends on the initial droplet size, decreasing with
a decrease in $r_0$.

**4.4.3 Effect of Ambient Temperature, $T_\infty$**

To determine the effect of a lower ambient temperature on droplet temperatures and lifetimes, Figs. 10 and 11
demonstrate similar plots as shown in Fig. 9, but for $T_\infty$ = 268.15 K (-5°C) and 263.15 K (-10°C), respectively. The
decrease in droplet temperatures and increase in droplet lifetimes depict similar relationships with $RH_\infty$ and $r_0$.
Droplets, irrespective of their initial size, cool to a lower temperature depending on the ambient $RH_\infty$, with the
magnitude of the cooling being inversely proportional to the subsaturation of the ambient environment. For instance,
for 10, 30 and 50 μm droplets, from an initial temperature of 268.15 K, the droplet temperatures approximately
decrease by 24 K, 11.4 K, and 4.7 K, for environments with $RH_\infty$ = 10%, 40%, and 70%, respectively (Table 1). The
droplet lifetimes for the 10 μm droplet are ~ 1.1 s, 1.8 s, and 3.9 s, while for the 30 μm droplet are ~ 11.4 s, 19.4 s,
and 42.8 s, and for the 50 μm droplet are ~ 32.8 s, 55.8 s, and 123.1 s, for $RH_\infty$ = 10%, 40% and 70%, respectively
(Table 2). Comparing these values with those of $T_\infty$ = 273.15 K (0°C), it can be noted that a lower ambient temperature
leads to a smaller decrease in droplet temperatures and a slight increase in droplet lifetimes in a spatiotemporally
evolving environment, for the same $RH_\infty$, $r_0$ and $P$, as shown by Roy et al., 2023. Fig. 11 and Table 1 depict that for
$T_\infty$ = 263.15 K (-10°C), the reduction in droplet temperatures is slightly smaller, ~ 21.8 K, 10.7 K, and 4.5 K for
environments with $RH_\infty$ = 10%, 40%, and 70%, respectively, and droplet lifetimes are longer relative to the higher
ambient temperatures of 273.15 K and 268.15 K (Table 2). This is because at a lower ambient temperature, the vapor
diffusivity into the ambient air is lower, leading to a weaker evaporation rate with slightly reduced cooling, and
extended droplet lifetime, relative to those in an environment with a higher ambient temperature.



Figure 9: Evolution of the decrease in temperature (in K, shaded contours) from the initial temperature of the domain = 273.15 K (0°C), and of the droplet radius (in μm, dashed black trace) for 10 (a,b,c), 30 (d,e,f), and 50 (g,h,i) μm droplets, immersed in an environment with $T_\infty$ = 273.15 K (0°C), $P$ = 500 hPa, and $RH_\infty$ = 10%, 40% and 70%.



| $T_\infty$ (K) | $r_0$ (µm) | $RH_\infty$ (%) | $P = 500$ hPa | | | | | $P = 850$ hPa | | | | |
|---|---|---|---|---|---|---|---|---|---|---|---|---|
| | | | $T_{WB\infty}$ (K) | $T_{RRD}$ (K) | $T_i$ (K) | $T_L$ (K) | $T_\infty - T_L$ (K) | $T_{WB\infty}$ (K) | $T_{RRD}$ (K) | $T_i$ (K) | $T_L$ (K) | $T_\infty - T_L$ (K) |
| 273.15 (0°C) | 10 | 10 | 264.94 | 264.06 | 264.15 | 247.15 | 26 | 267.20 | 266.49 | 266.35 | 249.03 | 24.12 |
| | | 40 | 267.95 | 267.41 | 267.35 | 261.09 | 12.06 | 269.30 | 268.85 | 268.95 | 261.40 | 11.75 |
| | | 70 | 270.67 | 270.43 | 270.35 | 268.21 | 4.94 | 271.28 | 271.07 | 271.10 | 268.29 | 4.86 |
| | 30 | 10 | 264.94 | 264.06 | 264.15 | 247.33 | 25.82 | 267.20 | 266.49 | 266.37 | 249.01 | 24.14 |
| | | 40 | 267.95 | 267.41 | 267.35 | 261.08 | 12.07 | 269.30 | 268.85 | 268.95 | 261.43 | 11.72 |
| | | 70 | 270.67 | 270.43 | 270.45 | 268.20 | 4.95 | 271.28 | 271.07 | 271.15 | 268.26 | 4.89 |
| | 50 | 10 | 264.94 | 264.06 | 264.15 | 247.31 | 25.84 | 267.20 | 266.49 | 266.37 | 249.04 | 24.11 |
| | | 40 | 267.95 | 267.41 | 267.36 | 261.09 | 12.06 | 269.30 | 268.85 | 268.95 | 261.45 | 11.7 |
| | | 70 | 270.67 | 270.43 | 270.45 | 268.20 | 4.95 | 271.28 | 271.07 | 271.15 | 268.29 | 4.86 |
| 268.15 (-5°C) | 10 | 10 | 261.64 | 260.90 | 260.98 | 244.12 | 24.03 | 263.57 | 263.01 | 263.15 | 246.32 | 21.83 |
| | | 40 | 263.96 | 263.50 | 263.48 | 256.77 | 11.38 | 265.16 | 264.79 | 264.82 | 257.17 | 10.98 |
| | | 70 | 266.13 | 265.91 | 265.9 | 263.47 | 4.68 | 266.68 | 266.51 | 266.65 | 263.57 | 4.58 |
| | 30 | 10 | 261.64 | 260.90 | 260.85 | 244.31 | 23.84 | 263.57 | 263.01 | 263.06 | 246.18 | 21.97 |
| | | 40 | 263.96 | 263.50 | 263.46 | 256.76 | 11.39 | 265.16 | 264.79 | 264.69 | 257.18 | 10.97 |
| | | 70 | 266.13 | 265.91 | 265.92 | 263.47 | 4.68 | 266.68 | 266.51 | 266.56 | 263.58 | 4.57 |
| | 50 | 10 | 261.64 | 260.90 | 260.85 | 244.29 | 23.86 | 263.57 | 263.01 | 263.06 | 246.21 | 21.94 |
| | | 40 | 263.96 | 263.50 | 263.47 | 256.76 | 11.39 | 265.16 | 264.79 | 264.72 | 257.16 | 10.99 |
| | | 70 | 266.13 | 265.91 | 265.92 | 263.46 | 4.69 | 266.68 | 266.51 | 266.56 | 263.56 | 4.59 |
| 263.15 (-10°C) | 10 | 10 | 258.14 | 257.55 | 257.53 | 241.38 | 21.77 | 259.73 | 259.28 | 259.28 | 243.49 | 19.66 |
| | | 40 | 259.89 | 259.51 | 259.65 | 252.46 | 10.69 | 260.90 | 260.60 | 260.65 | 252.97 | 10.18 |
| | | 70 | 261.56 | 261.38 | 261.4 | 258.73 | 4.42 | 262.04 | 261.90 | 261.90 | 258.88 | 4.27 |
| | 30 | 10 | 258.14 | 257.55 | 257.62 | 241.36 | 21.79 | 259.73 | 259.28 | 259.28 | 243.27 | 19.88 |
| | | 40 | 259.89 | 259.51 | 259.56 | 252.47 | 10.68 | 260.90 | 260.60 | 260.54 | 252.99 | 10.16 |
| | | 70 | 261.56 | 261.38 | 261.39 | 258.73 | 4.42 | 262.04 | 261.90 | 261.91 | 258.88 | 4.27 |
| | 50 | 10 | 258.14 | 257.55 | 257.62 | 241.37 | 21.78 | 259.73 | 259.28 | 259.28 | 243.48 | 19.67 |
| | | 40 | 259.89 | 259.51 | 259.56 | 252.47 | 10.68 | 260.90 | 260.60 | 260.56 | 252.99 | 10.16 |
| | | 70 | 261.56 | 261.38 | 261.39 | 258.73 | 4.42 | 262.04 | 261.90 | 261.91 | 258.87 | 4.28 |




**Table 1.** Comparison between thermodynamic wet bulb temperatures in the environment far away from the droplet ($T_{WB\infty}$), simulated droplet steady-state temperatures from Roy et al., (2023) ($T_{RRD}$), inflection point temperatures ($T_i$), and droplet temperatures at the end of their lifetimes from this study ($T_L$), in K, for initial droplet radii, $r_0$ = 10, 30 and 50 μm, relative humidities, $RH_\infty$ = 10, 40, 70%, and pressures, $P$ = 500 and 850 hPa, and ambient temperature, $T_\infty$ = 273.15 K (0°C), 268.15 K (-5°C) and 263.15 K (-10°C).



Figure 10: Same as Fig. 9 but for $T_\infty$ = 268.15 K (-5°C).







**Figure 11: Same as Fig. 9 but for $T_\infty = 263.15$ K (-10°C).**

### 4.4.4 Effect of Ambient Pressure, *P*

Figures. 12-14 depict the spatiotemporal evolution of the temperature and droplet radius similar to the previous figures, but now for a higher ambient pressure, $P = 850$ hPa, instead of 500 hPa as shown in Figs. 9-11. For a higher pressure, the corresponding decreases in droplet temperatures are smaller and droplet lifetimes are longer. Under the same environmental conditions but with an increase in ambient pressure, water vapor diffusivity decreases, leading to a decreased evaporation rate, reduced cooling, and extended droplet lifetimes. For example, for an environment with $T_\infty$





= 273.15 K (0°C), $P$ = 850 hPa (Fig. 12 and Table 1), 10, 30 and 50 µm droplets reach 249.0 K, 261.4 K, and 268.3
K for $RH_\infty$ = 10%, 40% and 70%, respectively, which are slightly higher as compared to the corresponding droplet
temperatures (247.3 K, 261.1 K, and 268.2 K) for $P$ = 500 hPa (Table 1). For higher ambient pressures, droplet
lifetimes are also increased due to reduced evaporation rate, with 50 µm droplet now surviving for 33.4 s, 55.8 s, and
121.7 s at $P$ = 850 hPa, instead of 27.4 s, 48.0 s, 107.5 s for $P$ = 500 hPa for $RH_\infty$ = 10%, 40% and 70%, respectively
(Table 2). Similar trends can also be observed for lower ambient temperatures, 268.15 K and 263.15 K, as shown in
Table 2, and Figs. 10 and 13, and 11 and 14.
**Figure 12: Same as Fig. 9 with $T_\infty$ = 273.15 K(0°C), but for $P$ = 850 hPa.**







**Figure 13: Same as Fig. 12 but for $T_\infty$ = 268.15 K (-5°C).**




**Figure 14: Same as Fig. 12 but for $T_\infty$ = 263.15 K (-10°C).**





| $T_\infty$ (K) | $r_0$ (μm) | $RH_\infty$ (%) | $P = 500$ hPa | | | | $P = 850$ hPa | | | |
|---|---|---|---|---|---|---|---|---|---|---|
| | | | $t_{LC}$ (s) | $t_{RRD}$ (s) | $t_L$ (s) | $t_L - t_{LC}$ (s) | $t_{LC}$ (s) | $t_{RRD}$ (s) | $t_L$ (s) | $t_L - t_{LC}$ (s) |
| 273.15 (0°C) | 10 | 10 | 0.26 | 0.56 | 0.87 | 0.61 | 0.44 | 0.77 | 1.11 | 0.67 |
| | | 40 | 0.39 | 0.89 | 1.51 | 1.12 | 0.66 | 1.18 | 1.79 | 1.13 |
| | | 70 | 0.78 | 1.86 | 3.36 | 3.36 | 1.33 | 2.43 | 3.87 | 2.54 |
| | 30 | 10 | 2.34 | 5.02 | 9.54 | 7.2 | 3.98 | 6.84 | 11.63 | 7.65 |
| | | 40 | 3.51 | 7.94 | 16.68 | 13.17 | 5.97 | 10.59 | 19.33 | 13.36 |
| | | 70 | 7.03 | 16.73 | 37.26 | 30.23 | 11.95 | 21.83 | 42.30 | 30.35 |
| | 50 | 10 | 6.51 | 13.95 | 27.43 | 20.92 | 11.06 | 19.06 | 33.35 | 22.29 |
| | | 40 | 9.76 | 22.08 | 48.04 | 38.28 | 16.59 | 29.45 | 55.78 | 39.19 |
| | | 70 | 19.52 | 46.46 | 107.45 | 87.93 | 33.18 | 60.64 | 121.70 | 88.52 |
| 268.15 (-5°C) | 10 | 10 | 0.38 | 0.72 | 1.05 | 0.67 | 0.65 | 1.01 | 1.32 | 0.67 |
| | | 40 | 0.58 | 1.12 | 1.77 | 1.19 | 0.98 | 1.54 | 2.15 | 1.17 |
| | | 70 | 1.15 | 2.31 | 3.91 | 2.76 | 1.96 | 3.14 | 4.60 | 2.64 |
| | 30 | 10 | 3.45 | 6.42 | 11.40 | 7.95 | 5.87 | 9.03 | 14.27 | 8.4 |
| | | 40 | 5.18 | 10.01 | 19.35 | 14.17 | 8.81 | 13.83 | 23.32 | 14.51 |
| | | 70 | 10.36 | 20.81 | 42.79 | 32.43 | 17.61 | 28.25 | 50.15 | 32.54 |
| | 50 | 10 | 9.59 | 17.88 | 32.76 | 23.17 | 16.31 | 25.15 | 40.99 | 24.68 |
| | | 40 | 14.39 | 27.86 | 55.76 | 41.37 | 24.46 | 38.48 | 67.02 | 42.56 |
| | | 70 | 28.78 | 57.80 | 123.10 | 94.32 | 48.92 | 78.48 | 144.07 | 95.15 |
| 263.15 (-10°C) | 10 | 10 | 0.57 | 0.95 | 1.29 | 0.72 | 0.98 | 1.37 | 1.68 | 0.7 |
| | | 40 | 0.86 | 1.45 | 2.13 | 1.27 | 1.47 | 2.08 | 2.68 | 1.21 |
| | | 70 | 1.72 | 2.98 | 4.60 | 2.88 | 2.93 | 4.21 | 5.66 | 2.73 |
| | 30 | 10 | 5.17 | 8.47 | 13.95 | 8.78 | 8.80 | 12.28 | 17.99 | 9.19 |
| | | 40 | 7.76 | 13.05 | 23.08 | 15.32 | 13.19 | 18.67 | 28.83 | 15.64 |
| | | 70 | 15.52 | 26.79 | 50.12 | 34.6 | 26.39 | 37.85 | 61.04 | 34.65 |
| | 50 | 10 | 14.37 | 23.59 | 40.11 | 25.74 | 24.43 | 34.19 | 51.59 | 27.16 |
| | | 40 | 21.56 | 36.30 | 66.42 | 44.86 | 36.65 | 51.93 | 82.53 | 45.88 |
| | | 70 | 43.12 | 74.43 | 144.33 | 101.21 | 73.30 | 105.16 | 175.50 | 102.2 |

**Table 2: Comparison between different timescales (in sec) in this and other studies, all for the cut off radii used in this**
**study. These include droplet lifetimes using the classical diffusion-limited evaporation approach ($t_{LC}$), the bulk droplet**
**approach in Roy et al., (2023) ($t_{RRD}$), and as calculated from this study ($t_L$), for initial droplet radii ($r_0$ = 10, 30 and 50 μm),**




relative humidities (*RH*∞ = 10, 40, 70%), and pressures (*P* = 500 and 850 hPa), and ambient temperature, *T*∞ = 273.15 K
(0°C), 268.15 K (-5°C) and 263.15 K (-10°C).

**5 Discussion**
**5.1 Droplet Temperature and Lifetime Comparison with Previous Studies**

As noted in the introduction, not many studies in the cloud microphysics literature have taken a close look at the
explicit numerical estimation of supercooled, evaporating cloud droplet temperatures for a wide range of
environmental conditions. Previously, a study by Srivastava and Coen (1992) investigated the evaporation of isolated,
stationary droplets by iteratively solving the steady-state solutions, using saturation vapor pressure relations from
Wexler (1976) to calculate saturation vapor density, and assumed the heat storage terms in the droplet heat budget to
be negligible. Solving for time-dependent heat and mass transfer between single, stationary cloud droplets evaporating
in infinitely large, prescribed ambient environments, Roy et al., (2023) demonstrated that the temperatures of the cloud
droplets reach steady-state quite quickly (< 0.5 s). Their steady-state droplet temperatures agreed well with those of
Srivastava and Coen (1992) and could be approximated by the thermodynamic wet-bulb temperature of the ambient
environment. In order to model a more realistic scenario of an isolated droplet evaporating in a subsaturated
environment, the current study advances the idealized framework of droplet evaporation as described in Roy et al.,
(2023) by including the impact of internal heat gradients within the droplet and resolving the spatiotemporally
evolving thermal and vapor density gradients between the droplet and its immediate environment to estimate the
evaporating droplet temperature and lifetime with higher accuracy.

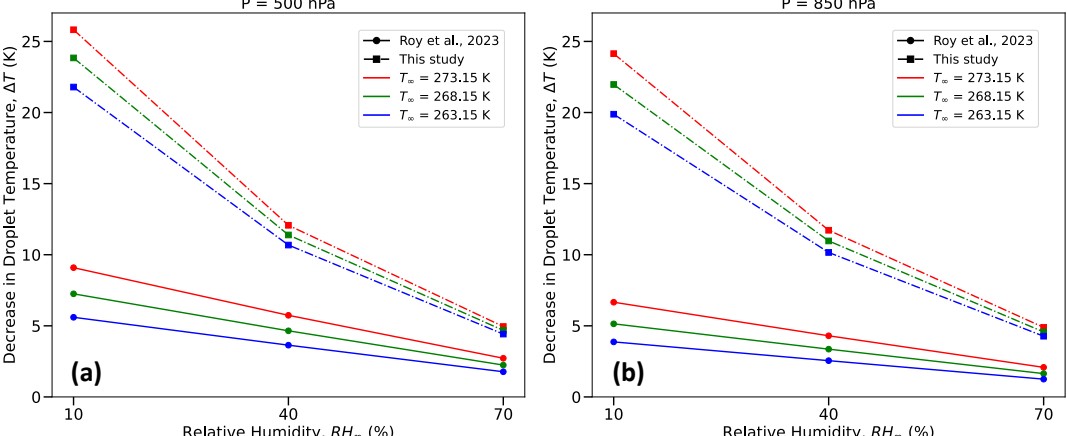

**Figure 15: Comparison between the decrease in droplet temperatures (in K) from an initial temperature the same as *T*∞,
calculated using the bulk droplet model from Roy et al., (2023) (dashed lines), and this study (dashed-dotted lines), for
initial droplet radii, *r₀* = 10, 30 or 50 μm, relative humidities (*RH*∞ = 10, 40, 70 %), and pressures, *P* = 500 hPa (left column),
and 850 hPa right column), and *T*∞ = 273.15 K (0°C, red), 268.15 K (-5°C, green) and 263.15 K (-10°C, blue).**
Table 1 provides a comparison between thermodynamic wet bulb temperatures of the initial environment ($T_{WB\infty}$),
simulated droplet steady-state temperatures from Roy et al. (2023) ($T_{RRD}$), and droplet temperatures at the end of their
lifetimes from this study ($T_L$), in K for several environments. Interestingly, the temperatures at the inflection point, $T_i$,





as defined in Sec. 3e, are in excellent agreement with $T_{WB\infty}$ and $T_{RRD}$. In the current study, the droplet temperature
continues to decrease almost steadily as the immediate environment in the vicinity of the droplet cools, finally reaching
$T_L$, unlike the evaporating droplet achieving steady-state temperature in a prescribed ambient environment far away
from the droplet in Roy et al., (2023). The evaporating droplet temperature essentially keeps adjusting to the
thermodynamic wet-bulb temperature of its immediate changing environment. Therefore, the more realistic
simulations of evaporating cloud droplets that include the effect of spatiotemporally varying ambient air thermal and
vapor density gradients, as shown in this study, reveal that droplets can potentially achieve even lower temperatures
than previously known or estimated from past studies (Srivastava and Coen, 1992; Roy et al., 2023). The decrease in
droplet temperatures from their initial temperatures can be much larger, especially for drier environments, as much as
25.8 K for $RH_\infty$ = 10% and 5.0 K for $RH_\infty$ = 70%, for an environment with $P$ = 500 hPa, and $T_\infty$ = 273.15 K (Table 1
and Fig. 15a). As shown in Fig. 15, the magnitude of reduction in droplet temperatures decreases with higher ambient
$RH_\infty$ and $P$, and lower $T_\infty$, similar to previous studies (Srivastava and Coen, 1992; Roy et al., 2023).

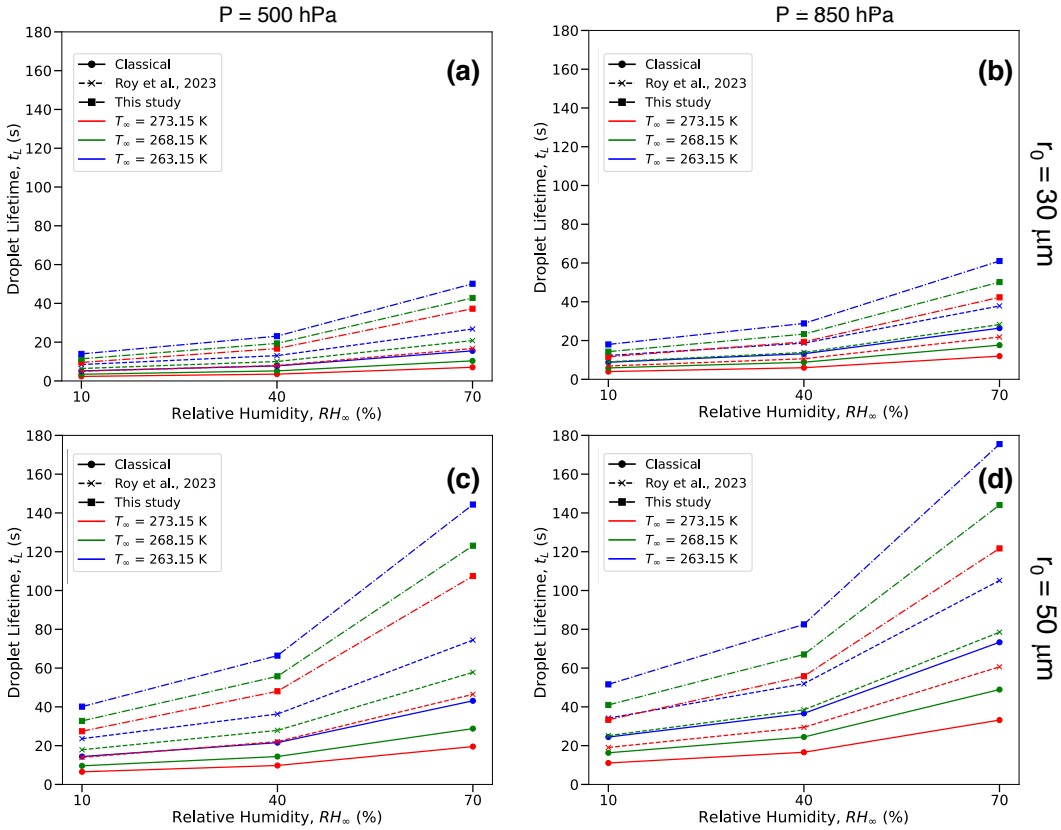

**Figure 16: Comparison between droplet lifetimes (as defined in this study) calculated using the classical diffusion-limited**
**evaporation approach (solid lines), bulk droplet model from Roy et al., (2023) (dashed lines), and this study (dashed-dotted**
**lines), for initial droplet radii, $r_0$ = 30 μm (upper panel), and 50 μm (lower panel), relative humidities ($RH_\infty$ = 10, 40, 70%),**
**and pressures, $P$ = 500 hPa (left column), and 850 hPa right column), and ambient temperature, $T_\infty$ = 273.15 K (0°C, red),**
**268.15 K (-5°C, green) and 263.15 K (-10°C, blue). 10 μm droplets (not shown here) have much smaller lifetimes compared**
**to 30 and 50 μm droplets.**



Table 2 and Fig. 16 provide comparisons between 30 and 50 µm droplet lifetimes (as defined earlier in Sec. 3a) using
the classical pure-diffusion-limited evaporation approach $(t_{LC})$, which ignores evaporative cooling at the droplet
surface (Maxwell, 1890; Eq 13-10 of Pruppacher and Klett, 1997), the "bulk" droplet approach as described in Roy et
al., (2023) $(t_{RRD})$, which ignores internal droplet heat transfer and spatiotemporally varying thermal and moisture
gradients in the ambient air, and results from this study $(t_L)$. The magnitude of $t_L$ is greater than the corresponding
values of $t_{LC}$ and $t_{RRD}$. This is because the droplet temperatures in this study never reach steady-state and are much
lower than the corresponding droplet temperatures from the diffusion-limited approach ($\sim T_\infty$), and Roy et al., (2023)
($\sim T_{RRD}$). This can be explained by the greater decrease in evaporating droplet temperature leading to a greater reduction
in saturation vapor pressure at the droplet surface. This results in a slower droplet evaporation rate, therefore increasing
the droplet lifetime. As shown in Fig. 16, this increase in droplet lifetime depends on the environmental subsaturation,
ambient temperatures, and pressures, with a greater increase for more humid, higher pressure, and lower ambient
temperature environments. This increase in droplet lifetimes can potentially enhance ice nucleation by increasing the
chances of activation of ice nucleating particles (INPs) within the supercooled cloud droplets (see Section 5b).

**5.2 Implications for ice nucleation**

Ice nucleation rates are influenced by temperature (Wright and Petters, 2013; Kanji et al., 2017) and time (Vali, 1994).
There are two theories in ice nucleation modeling: the time-independent "singular hypothesis," which suggests
instantaneous ice formation, and the time-dependent "stochastic hypothesis," which proposes that ice clusters in
embryos form and vanish continually, with a frequency that depends on temperature. Supercooled cloud droplet
temperatures and their lifetimes are potential contributing factors for the enhancement of ice formation within
evaporating regions of clouds such as cloud-tops and edges. As discussed in Roy et al., (2023), evaporative cooling
of supercooled cloud droplets in subsaturated environments can enhance ice nucleation near cloud boundaries in two
ways: by instantly increasing ice-nucleating particle activation due to lower droplet temperatures (consistent with the
singular hypothesis) and/or by extending supercooled droplet lifetimes, allowing more time for nucleation events
(consistent with the stochastic hypothesis). Based on limited laboratory investigations available on time dependency
of heterogeneous ice nucleation, conducted between temperatures -14 and -30 °C, varying fractions of the droplets
were reported to freeze within a range of 1 s to 500 s (Welti et al., 2012; Broadley et al., 2012; Murray et al., 2012;
Jakobsson et al., 2022). As shown in Table 2 and Fig. 16, droplet lifetimes as estimated from both approaches ($t_{RRD}$
and $t_L$), which include droplet evaporative cooling, are longer as compared to the classical diffusion-limited
evaporation approach ($t_{LC}$), allowing more time for potential occurrence of an ice nucleation event. For temperatures
between -5 °C and -10 °C, for the three different subsaturated environments ($RH_\infty$ = 10, 40, and 70%) examined in
this analysis, $t_{RRD}$ typically ranged from 0.7-4.2 s for 10 µm, 6-38 s for 30 µm and 18-105 s for 50 µm initial radius
of droplets, respectively. For similar environments, $t_L > t_{RRD} > t_C$, with $t_L$ typically ranging from 1.1-5.7 s for 10 µm,
11-61 s for 30 µm and 33-176 s for 50 µm initial radii droplets, respectively. For larger droplets, say 30 and 50 µm,
the droplets survive much longer as compared to 10 µm droplets, likely enhancing the chances of an ice nucleation
event. Comparing these values with reported droplet freezing timescales available from experimental studies, droplet



freezing events can potentially occur within the time frame when these droplets can reach lower temperatures due to
evaporative cooling before they completely dissipate into the subsaturated air. Results from this study further
strengthen evidence of the hypothesized mechanism of enhancement of ice nucleation via droplet evaporation.
Together with the consistent observation of supercooled water in cloud-top generating cells (Plummer et al., 2014;
Zaremba et al., 2024), these results contribute to explaining the observations of the prodigious production of ice
particles produced in generating cells at the cloud-tops of winter storms (e.g., Plummer et al., 2015).
Due to the observational evidence of a higher dependency of ice nucleation on temperature than time (Wright and
Petters, 2013), and the increased difficulty of representing time-dependent stochastic nucleation in numerical models,
the simpler and more widely used approach is to use the time-dependent singular hypothesis framework to simulate
ice initiation processes. Drawing from theoretical insights, laboratory experiments, and field campaigns, numerous
parameterization methods for modeling heterogeneous ice nucleation in cloud and climate models have been created
over the years (Fletcher, 1962; Cooper, 1986; Meyers et al., 1992; DeMott et al., 1998; Khvorostyanov and Curry,
2000; Phillips et al., 2008). Most of the conventionally used schemes (Fletcher, 1962; Cooper, 1986; Demott et al.,
2010) share a common feature, which is the utilization of the ambient air temperature for estimating activated INPs,
as opposed to relying on the droplet temperature, even for primary ice-nucleation modes such as immersion freezing
and contact nucleation.
Similar to Roy et al. (2023), we investigate the maximum enhancement in activated INP concentrations that can occur
due to evaporative cooling of supercooled water droplets in a spatiotemporally varying environment, assuming that
the activation in the parameterization schemes (Fletcher, 1962; Cooper, 1986; Demott et al., 2010) is related to the
droplet temperatures towards the end of their lifetimes ($T_L$) rather than the ambient temperature. Fig. 17 presents a
comparison between Roy et al. (2023), and the current study in terms of the highest fractional increase in activated
ice-nucleating particles (INPs), as projected through the Fletcher, Cooper, and Demott schemes (considering ambient
aerosol concentration, $N_a$, with diameters greater than 0.5 µm). Owing to even lower droplet temperatures during
evaporation, the fractional increase in activated INPs is higher as calculated from this study, with several orders of
magnitude increase for drier environments. For example, the Fletcher Scheme predicts an enhancement in activated
INPs by a factor of ~$10^6$ for $RH_\infty = 10\%$, $T_\infty = 268.15$ K, $P = 500$ hPa based on droplet temperatures from this study,
while the corresponding number from Roy et al. (2023) is ~100 (Fig. 17a). The fractional increases are slightly smaller
for higher pressure environments due to lower evaporative cooling of the droplets under such conditions (compare
Figs. 17a, d, b,e, and c,f). Consistent with previous results from Roy et al. (2023), compared to the Fletcher Scheme,
the Cooper and Demott schemes demonstrate relatively lower enhancement in activated INPs. For the same
environment stated earlier, the corresponding activated INP enhancement factor values for Cooper and Demott
schemes are ~$10^3$ and 80, respectively (Figs. 17b and c).
Therefore, results from the current study further corroborate the hypothesized ice nucleation enhancement mechanism
through evaporative cooling of supercooled droplets (Mossop et al., 1968; Young, 1974; Beard, 1992; Roy et al.,



2023), providing much higher estimates of activated INP concentrations from previous analyses (Roy et al., 2023).
This potential increase in INP concentrations in subsaturated environments near cloud tops and edges, particularly at
higher sub-freezing temperatures, may partially help resolve the several orders of magnitude discrepancy between
predicted INP and observed ice particle concentrations in such regions of the cloud. To evaluate the effectiveness of
the potential ice-nucleation enhancement mechanism through evaporation, future modeling experiments within a
robust dynamical model setup, considering a population of both freezing and evaporating droplets, along with their
lifetimes, droplet-droplet interaction, different species of INPs, impact of turbulence and other feedbacks, are required.

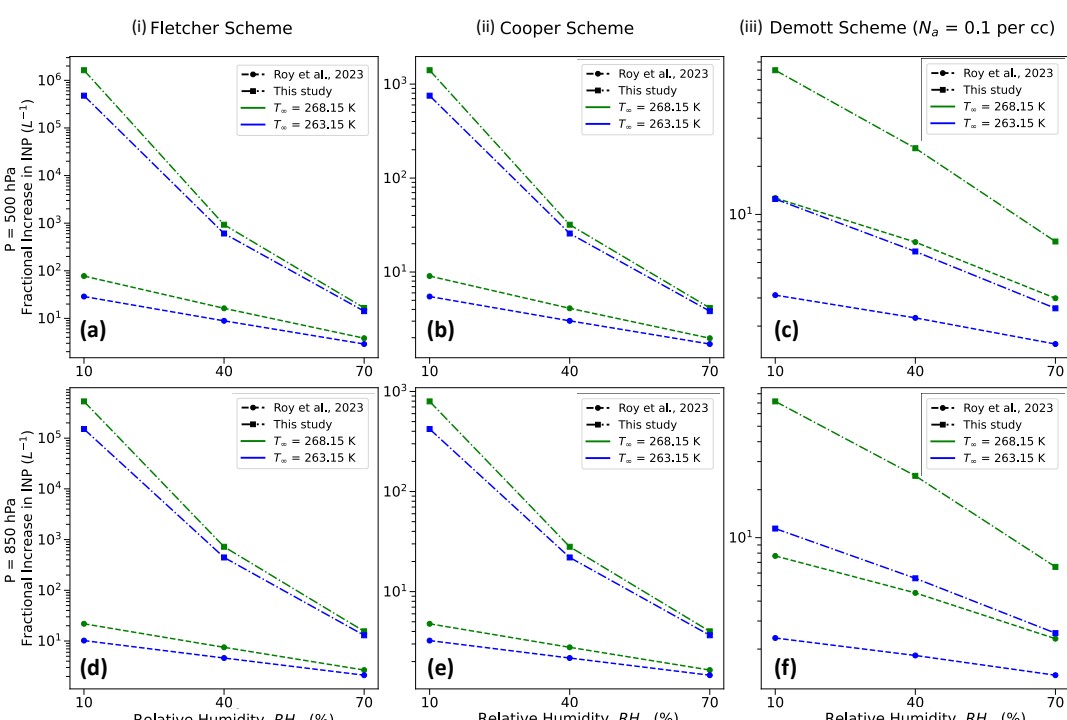

**Figure 17: Comparison between the maximum fractional increase in INPs as estimated by Roy et al., (2023) and this study**
**for three different parameterization schemes: (i) Fletcher (1962) (ii) Cooper (1986), and (iii) Demott et al., (2010), for three**
**different environmental relative humidities ($RH_\infty$ = 10, 40 and 70%), and two ambient temperatures ($T_\infty$ = 268.15 K (-5°C)**
**and 263.15K (-10°C)) and two different pressures ($P$ = 500 and 850 hPa).**

**6 Conclusions**

In this study, we presented a quantitative investigation the temperature and lifetime of an evaporating droplet,
considering internal thermal gradients within the droplet as well as resolving thermal and vapor density gradients in
the surrounding ambient air. The computational approach involved solving the Navier-Stokes and continuity
equations, coupled with heat and vapor diffusion equations, using an advanced numerical model that employs the
finite element method. This is the first simulation of the spatiotemporal evolution of droplet temperature, radius, and



its environment for an isolated, stationary, and supercooled cloud droplet evaporating in various subsaturated
environmental conditions. Various ambient pressure ($P$), temperature ($T_\infty$), relative humidity ($RH_\infty$), and initial droplet
radii ($r_0$) were considered. The motivation behind this study was to provide more exacting calculations to support the
hypothesized ice nucleation enhancement mechanism due to the evaporation of supercooled cloud droplets at cloud
boundaries, such as cloud-top ice-generating cells, and for ambient temperatures between 0°C and -10°C where ice
nucleation is least effective.

Since evaporation is a surface phenomenon, there is a legitimate interest in computing droplet internal thermal
gradients and investigating if the droplet surface gets preferentially cooled during droplet evaporation, with regards to
the activation of ice nucleating particles. The numerical simulations show for typical cloud droplet sizes ($r_0 = 10, 30,$
50 μm) and environmental conditions considered here, the internal thermal gradients dissipate quite quickly ($\leq 0.3$ s)
when the droplet is introduced to a new subsaturated environment. Thus, spatial thermal gradients within the droplet
can be reasonably ignored. Hence, one can potentially ignore the extra computational expense of simulating
conductive heat transfer within the droplet for timescales > 1 s.

The results from this study support findings from the literature that an evaporating supercooled cloud droplet can exist
at a temperature lower than that of the ambient atmosphere and corroborate the tendencies of the dependence of
decrease in droplet temperatures on environmental factors and initial droplet sizes (Srivastava and Coen, 1992; Roy
et. al, 2023). Decreases in droplet temperatures are smaller for higher ambient $RH_\infty$ and $P$, and lower $T_\infty$, qualitatively
in accordance with previous studies (Srivastava and Coen, 1992; Roy et al., 2023). The novelty of this study lies in
demonstrating that the magnitude of droplet cooling can be much higher than estimated from past studies of droplet
evaporation, especially for drier environments. For a droplet evaporating in an environment with $P = 500$ hPa, $T_\infty =$
268.15 K (-5°C), $RH_\infty = 10\%$, Roy et al., (2023) estimated a 7.3 K decrease in droplet temperature, while this study
shows that there can be as much as a 23.8 K decrease in droplet temperature. This is because previous studies assumed
prescribed ambient environments at all distances from the droplet, while this analysis shows that as a droplet
evaporates and cools, the air in the vicinity of the droplet cools as well, giving rise to spatiotemporally varying thermal
and vapor density fields in the immediate environment surrounding the droplet. Here, the net conductive warming
from the environmental air enveloping the droplet is lower as compared to Roy et al., (2023), effectively leading to a
much lower droplet temperature. At a particular time, the strength and radial dependence of these gradients depend on
the subsaturation of the air medium and the magnitude of droplet cooling due to evaporation, with the largest cooling
at lower $RH_\infty$. In this study, the temperature and vapor density in the ambient air continually evolve, thus affecting the
transfer of heat and vapor between the droplet surface and the environment far away from the droplet. This affects the
temperature evolution and decay rates of the evaporating droplet to a greater degree than shown in previous studies
for a similar environment (Srivastava and Cohen, 1992; Roy et al. 2023).

This study also demonstrated that the lifetimes of the evaporating droplets are longer compared to Roy et al. (2023)
because as the droplet temperature gets lower, the saturation vapor pressure at the droplet surface reduces, leading to



a weaker evaporation rate. For an environment with $P$ = 500 hPa, $T_\infty$ = 268.15 K (-5°C), $RH_\infty$ = 10%, a 50 μm droplet
reaches the end of its lifetime, as defined in this study, in 32.8s, while the corresponding values for the diffusion-
limited evaporation approach as estimated from Roy et. al, (2023) are 9.6 s and 17.9 s, respectively. The rates of
evaporation tend to be lower in this study due to even lower droplet temperatures as well as spatiotemporally varying
vapor density gradients around the droplets. As the droplet evaporates, the envelope of air surrounding the droplet is
colder, has lower values of diffusivity leading to lower evaporation rates, and has higher vapor concentration than the
ambient air, thus decreasing the evaporation rates.

To summarize, if one considers the more realistic case of droplet evaporation, including the spatiotemporally varying
thermal and vapor density gradients in the vicinity of the water droplet, the evaporating droplet can experience a
substantial reduction in temperatures by tens of degrees, strongly dependent on the ambient relative humidity and
weakly dependent on ambient pressure and temperature. Similar to the case of an isolated, stationary droplet
evaporating in a prescribed ambient environment, the droplet almost immediately reaches its inflection point
temperature, which can be well-approximated by the thermodynamic wet-bulb temperature of the initial ambient
environment around the droplet. However, unlike the former case, the droplet temperatures in this study continue to
steadily decrease as they adjust to the evolving thermodynamic wet-bulb temperature of the surrounding air. In more
humid environments, the droplets may not experience a larger droplet cooling, but their lifetimes, as defined in this
study, get extended by tens of seconds as compared to the classical estimation which neglects droplet cooling.

The current analysis also demonstrates that lower evaporating droplet temperatures would lead to an enhancement of
activated INPs from three widely used INP parameterization schemes, further corroborating the hypothesized ice
nucleation enhancement mechanism through evaporative cooling of supercooled droplets. Notably, the estimates of
activated INP concentrations from this study are higher than previous analyses, as the droplet temperatures are much
lower towards the end of their lifetimes, with several orders of magnitude increase in activated INPs for drier
environments. The Fletcher Scheme predicts the greatest enhancement in activated INPs by a factor of ~$10^6$ for $RH_\infty$
= 10%, $T_\infty$ = 268.15 K, P = 500 hPa, while the corresponding enhancement factor values for Cooper and Demott
schemes are ~$10^3$ and 80, respectively.

This study suggests a need for a more in-depth examination of supercooled cloud droplet temperatures and their
lifetimes in subsaturated environments, especially when simulating heterogeneous ice nucleation processes that
require the presence of supercooled water droplets. This is crucial because the concentration of activated ice-
nucleating particles (INPs) is influenced by both droplet temperature and how long evaporating droplets persist.
Additionally, the findings from this investigation may also partially help understand disparities between observed ice
particle concentrations and activated INPs, especially at relatively higher sub-0°C temperatures. Including the effect
of droplet evaporative cooling on droplet temperatures and lifetimes, while modeling cloud microphysical processes
in subsaturated environments, will also lead to improved accuracy of the evolution of the droplet size distribution as
well as primary ice nucleation mechanisms.



**Author contribution:** PR, RMR and LDG conceptualized the problem and numerical experiments. PR designed and
performed the simulations, analyzed the data, and prepared the first draft of the manuscript. RMR and LDG reviewed
and edited the manuscript. RMR and LDG acquired required funding for the project.

**Competing interests:** The authors have no competing interests.

**Acknowledgements:** This work was funded by the NASA CAMP²Ex program under grant 80NSSC18K0144 and the
NASA Earth Venture Suborbital-3 (EVS-3) IMPACTS program under grant 80NSSC19K0355. This research was
also supported by the National Science Foundation under grant NSF AGS-2016106.

**Code/Data availability:** This modeling analysis used the proprietary COMSOL Multiphysics version 6.0 software
package which can be licensed through https://www.comsol.com/.

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
