# Peer review of "Evolution of Cloud Droplet Temperature and Lifetime in"

_EGUsphere, 2024_

## Referee Comment (RC1)

**Review on article egusphere-2024-526 with title "Evolution of Cloud Droplet Temperature and Lifetime in Spatiotemporally Varying Subsaturated Environments with Implications for Ice Nucleation at Cloud Edges"**

This paper presents a study of the transient effects on the droplet temperature and lifetime when the droplet is suddenly immersed in a drier environment. The motivation is that this transient has been hypothesized to play an important role in ice nucleation. The authors find that the temperature reduction can be threefold the values found in the steady solution.

The paper could be an interesting contribution to the topic of ice micro-physics. However, some parts of the setup, the results and the explanations were a bit unclear. I provide the details below.

1. I am not sure about the novelty of the result that the gradients inside the droplet dissipate quickly. The standard estimate of a diffusive time is $r_0^2/\kappa$, where $r_0$ is the length scale, in this case the droplet's radius, and $\kappa$ is the thermal diffusivity of water. For the typical values consider in this paper, one find time scales which are indeed less that $0.1\ s$. What would be the new contribution from the detailed simulations in this paper for this particular aspect of the gradients inside the droplet?

2. What is the physical mechanism that explains the large difference of the droplet's temperature with respect to the steady solution. I understand that the rapid evaporation cools the droplet quickly, but the water vapor needs to be diffused away for the evaporation to continue, and thermal energy is diffused similarly quick towards the droplet. Textbooks like Rogers and Yau [1989] and Lamb and Verlinde [2011] retain the effect of the gradients and the evaporative cooling. Normally, the effect on temperature is small but the effect on evaporative rates is large because of the nonlinearity of Clausius-Clapeyron, which why I was surprised to see the large changes in $T$. Why is the diffusion of thermal energy towards the droplet not compensating the evaporative cooling as efficiently as in the steady state case?

   The authors refer to the dependence of the vapor diffusion coefficient on $T$ to explain part of this behavior, e.g. in line 769 in the conclusions. However, this dependence accounts for a change of $D$ of only $\approx 7-8\%$, whereas the temperature difference between the droplet and the environment is 2-3 times, and this should lead to larger thermal energy flux towards the droplet.

3. In figure 4.2, it seems that the droplet's temperature never reaches a steady state. However, since the environment has fixed conditions, it should eventually reach a steady state, provided that the droplet does not evaporate before. How long does it take to reach this steady state in the cases where the droplet has not evaporated yet by the end of the

simulation? Somehow it looks like the thermal energy in the droplet is decoupled from the environment and cannot be warmed.

4. I was confused by the set-up of the problem. It seems that the authors use cylindrical coordinates instead of spherical coordinates. At the beginning of 3.1., I understood that still the droplet is spherical and it was only the overall domain that was cylindrical, which is fine. But then section 3.4 indicates that boundary conditions are applied at the center of the domain $r = 0$ and not at the center of the droplet, which would mean a whole diameter across the droplet. That was confusing because it seems to indicate that the authors do not consider a spherical droplet but a cylindrical droplet. Is this so? If so, what are the implications in the results? Maybe the authors can clarify this in the paper.

5. In section 3.2, the equations do not have a number, which makes it difficult to refer to them.

6. In section 3.2, the mass diffusivity retains the dependence on $T$ but the other molecular transport coefficients dropt that dependence. Why? If the dependence of $D$ on $T$ plays a role, at least the dependence of the thermal conductivity $k$ on $T$ should be explored, since both transport phenomena are equally important for the problem. Maybe the effect of $T$ on $k$ is negligible compared to that of $T$ on $k$, but then the authors could explain it.

7. In section 3.2, what is the mathematical expression to calculate $Q_b$?

8. In section 3.2, a reference for the interfacial conditions for the stresses would be useful.

9. In line 392, the authors use the term "the inflection point in the curves," but this might be misleading because those are not inflection points in the usual sense of calculus and curves, i.e., change in the sign of the curvature. Maybe a different term is convenient.

10. In section 4, most of the text repeats what one can see in the figures without adding more insight about the reasons behind that behavior. It might be more helpful to discuss more the physics behind the results.

11. For instance, in section 4.2, the dependence on $T_\infty$ and $P$ seems to be much smaller than that on $RH_\infty$. Why? It might be more useful to differentiate those dependences and concentrate on the latter one instead of showing all figures. Otherwise, there are so many figures that it becomes difficult to distilled the new information that they convey.

12. In line 431, the authors write "while the decrease is 5 $K$ in 120 $s$." However, figure 6 seems to indicate 2 $K$ instead of 5 $K$. Is the reference to the figure correct?

13. In that same section, section 4.3, it seems that the main result is that the droplet's temperature is well approximate by the wet-bulb temperature. On the other hand, would not that be expected because of the definition of wet-bulb temperature?

    Also, it is difficult to see something in Fig 8. Why not plot the wet-bulb temperature in figure 4 and 5 and see how it approaches the droplet's temperature at a time $T_i$?

14. Line 489 indicates the "The droplet lifetimes vary widely...", but I am not sure what is meant by widely because the lifetime changes by a factor of 4 while the RH has been changed by a factor of 7. It seems that the lifetime simply follows the change in the control parameter RH.

15. Line 496 says "From Figs. 9-14, the decrease in droplet temperatures is independent of the initial droplet size if all other initial environmental conditions are kept constant.". I am not sure if I understand this because fig 9 shows a difference between panels a, d, an g. In fig 4 and 5, the initial decrease of $T$ until $T_i$ seems independent of $r_0$ but later on there seems to be a dependence...

16. In line 512, "The decrease in droplet temperatures and increase in droplet lifetimes depict similar relationships with $RH_\infty$ and $r_0$." Why is the dependence on $T_\infty$ smaller than the dependence on other parameters?

17. Line 634 indicates "the classical pure-diffusion-limited evaporation approach, which ignores evaporative cooling at the droplet surface". I am not sure what is meant because this effect is discussed textbooks like Rogers and Yau [1989] and Lamb and Verlinde [2011] which are often used to teach microphysics.

18. Line 669 seems to contain the main message "Comparing these values with reported droplet freezing timescales available from experimental studies, droplet freezing events can potentially occur within the time frame when these droplets can reach lower temperatures due to evaporative cooling before they completely dissipate into the subsaturated air."

    However, this message was a bit lost in the middle of a long paragraph. Maybe it helps to make the statement at the beginning and then elaborate it.

19. In the conclusions, the first sentence says "...internal thermal gradients within the droplet as well as resolving thermal and vapor density gradients in the surrounding ambient air." suggesting that this is new.

    However, this gradients are retained in the classical studies, e.g., Rogers and Yau [1989] and Lamb and Verlinde [2011].

    Do the authors mean the temporal variation of those gradients?

**References**

D. Lamb and J. Verlinde. *Physics and Chemistry of Clouds*. Cambridge University Press, 2011.

R. R. Rogers and M. K. Yau. *A Short Course in Cloud Physics*. Butterworth-Heinemann, third edition, 1989.

---

## Author Comment (AC1)

**Response to Reviewer 1**

We thank the reviewer for providing insightful comments and suggestions on our paper. We believe that responding to the comments has further improved the quality of the manuscript. We have addressed the comments below. The reviewer comments are in **black**, our responses in **red** and any text changes in **blue**.

**Review 1**

This paper presents a study of the transient effects on the droplet temperature and lifetime when the droplet is suddenly immersed in a drier environment. The motivation is that this transient has been hypothesized to play an important role in ice nucleation. The authors find that the temperature reduction can be threefold the values found in the steady solution.

The paper could be an interesting contribution to the topic of ice micro-physics. However, some parts of the setup, the results and the explanations were a bit unclear. I provide the details below.

1. I am not sure about the novelty of the result that the gradients inside the droplet dissipate quickly. The standard estimate of a diffusive time is $r_0^2/\kappa$, where $r_0$ is the length scale, in this case the droplet's radius, and $\kappa$ is the thermal diffusivity of water. For the typical values consider in this paper, one find time scales which are indeed less that $0.1\ s$. What would be the new contribution from the detailed simulations in this paper for this particular aspect of the gradients inside the droplet?

   The reviewer's standard estimate is correct and is consistent with the results shown in the original Figure 3, whose point was to simply show that diffusive timescales are a fraction of a second. To shorten the manuscript and address the reviewer's main point, we have removed the original Fig. 3 and reduced the discussion of the internal temperature gradient dissipation within the droplet, noting that the model results conform to the diffusive timescale as pointed out by the reviewer.

   Comments #2 and #11 are related in that they ask for more insight into the physical mechanisms regarding droplet evaporative cooling. We organize the response below to answer these two questions together.

2. What is the physical mechanism that explains the large difference of the droplet's temperature with respect to the steady solution. I understand that the rapid evaporation cools the droplet quickly, but the water vapor needs to be diffused away for the evaporation to continue, and thermal energy is diffused similarly quick towards the droplet. Textbooks like Rogers and Yau [1989] and Lamb and Verlinde [2011] retain the effect of the gradients and the evaporative cooling. Normally, the effect on temperature is small but the effect on evaporative rates is large because of the nonlinearity of Clausius-Clapeyron, which why I was surprised to see the large changes in $T$. Why is the diffusion of thermal energy towards the droplet not compensating the evaporative cooling as efficiently as in the steady state case?

   The authors refer to the dependence of the vapor diffusion coefficient on $T$ to explain part of this behavior, e.g. in line 769 in the conclusions. However, this dependence accounts for a change of $D$ of only $\approx 7-8\%$, whereas the temperature difference between the droplet and the environment is 2-3 times, and this should lead to larger thermal energy flux towards the droplet.

11.        For instance, in section 4.2, the dependence on $T_\infty$ and $P$ seems to be much smaller than that on $RH_\infty$. Why? It might be more useful to differentiate those dependences and concentrate on the latter one instead of showing all figures. Otherwise, there are so many figures that it becomes difficult to distilled the new information that they convey.

The dependence on $T_\infty$ and $P$ is much smaller than that on $RH_\infty$. The strong dependence on $RH_\infty$ compared to temperature results from the initial conditions. The droplet temperature initially is in thermal equilibrium with its environment (the droplet has the same temperature as that of the far environment), but the vapor field is far from equilibrium, especially for low relative humidity environments. As a result, the vapor diffusion rate (which depends on the vapor density gradient) far exceeds the thermal diffusion rate (which depends on the temperature gradient). Because the cloud droplets are small, and the relative humidity gradients are large, the droplets never come to an equilibrium state before evaporating completely into the subsaturated air. The water vapor flux into the larger subsaturated environment maintains a vapor density near the droplet surface that approaches but never reaches saturation. As a result, the wet-bulb temperature near the droplet surface continues to fall but at a slower rate that depends on $RH_\infty$ (Fig. 11). The pressure affects both the moisture and temperature diffusion fluxes, so these scale with each other, resulting in pressure not having a stronger effect compared to that of the moisture gradient. We added the following paragraph to the Discussion section (new Section 5.1) explaining the relative dependencies.

Figure 11 shows the evolution of the droplet surface temperature and the thermodynamic wet-bulb temperature in the air at the grid point nearest to the droplet surface for a range of conditions. Initially the droplet temperature rapidly decreases to the thermodynamic wet-bulb temperature of the far environment. The novel result from this study is that the droplet temperature continues to decrease because of the non-equilibrium condition of the thermal and vapor fields during the evaporation process. The droplet temperature continues to conform to the wet-bulb temperature directly adjacent to the droplet surface, which is lower than the wet-bulb temperature of the far environment. Note that the dependence on $T_\infty$ and $P$ is much smaller than that on $RH_\infty$. The strong dependence on $RH_\infty$ compared to temperature results from the initial conditions. The droplet temperature initially is in thermal equilibrium with the far environment, but the vapor field is far from equilibrium, especially for low relative humidity environments. As a result, the vapor diffusion rate (which depends on the vapor density gradient) far exceeds the thermal diffusion rate (which depends on the temperature gradient). Because the cloud droplets are small, and the relative humidity gradients are large, the droplets never come to an equilibrium state before evaporating completely into the subsaturated air. The water vapor flux into the larger subsaturated environment maintains a vapor density near the droplet surface that approaches but never reaches saturation. As a result, the wet-bulb temperature near the droplet surface continues to fall but at a slower rate that depends on $RH_\infty$ (Fig. 11). The pressure affects both the moisture and temperature diffusion fluxes, so these scale with each other, resulting in pressure not having a stronger effect compared to that of the moisture gradient.

We reexamined the Rogers and Yau [1989], Pruppacher and Klett (1997), and Lamb and Verlinde [2011] textbooks. None of these texts discuss the problem of cloud droplet evaporation explicitly. Pruppacher and Klett (1997, page 537-546) provide the steady state equations that can be used to calculate falling rain drop evaporation, but never address the problem of cloud droplet evaporation and its effect on temperature. For cloud droplets, these texts focus on condensational growth. Based on Pruppacher and Klett (1997), typical supersaturations in cumulus clouds rarely exceed 1%. Under these levels of supersaturation, the steady state approach is quite valid. The equivalent subsaturations for droplet evaporation would be relative humidity (RH) of ~99%. We ran additional simulations which we are not including in the paper but show here for RH = 99% to examine whether the droplet

temperatures approach steady state close to ambient temperature of 268.15 K used in these simulations. Results are shown in the figure below. The figure shows that the temperature of the 50 μm droplet initially decreases by 0.072 K to reach the ambient wet bulb temperature and then over the next 1000 seconds of evaporation, the decrease in droplet temperature is only 0.021 K.

Therefore, to a good approximation, the difference between the ambient and droplet temperatures is sufficiently small that the steady state approximation (e.g. Eq. 8.15 of Lamb and Verlinde [2011]) can be used with little error in the droplet evaporation equation. In the simulations in this paper which are at a much lower relative humidities, steady state conditions are no longer applicable.

[Figure]

The process we are simulating starts in a condition that is far from equilibrium and undergoes a time evolution as it attempts to evolve towards a steady-state where the thermal energy towards the droplet compensates for evaporative cooling. The droplet initially rapidly cools to the thermodynamic wet-bulb temperature of the environment similar to what has been shown in Roy et al., 2023. What is novel is that under low relative humidity conditions, the thermal and vapor diffusion are still not in equilibrium. As the system attempts to achieve equilibrium, the combination of vapor and thermal diffusion in the immediate vicinity of the drop leads to a gradual reduction in the wet-bulb temperature of the immediate droplet environment leading to a continued slow decrease in the droplet temperature as the droplet continues to evaporate. Under high RH (say RH > 90%) conditions, as in the simulations we show above, the equilibrium is established soon, and the droplet comes to a steady-state temperature approximately equal to the initial thermodynamic wet bulb temperature of the environment.

The simulations in the current paper deal with RH values characteristic of the dry environments typically found just above cloud tops of stratiform clouds where cloud-top generating cells are often observed. These RH values above cloud top typically vary from as low as 10% to as high as 70%. Under conditions where air has a history of descent above a stratiform cloud deck, the vertical humidity gradients can be quite large. Observations of relative humidity vertical profiles above Arctic stratocumulus clouds reveal strong vertical gradients in RH at cloud top, with RH values as dry as ~ 50% (Egerer et al., 2021). Also, during the SNOWIE field campaign, dry layer incursions (with RH ~10-20%) were observed above orographic clouds, which led to very sharp gradients in RH at cloud

tops (Xue et al. 2022). Under such conditions, the cloud droplets exposed to these higher subsaturations are far from equilibrium. In this study, the numerical simulations conducted to model the explicit vapor and thermal diffusion coupling between the evaporating cloud droplet and its near environment demonstrate that the droplet never reaches equilibrium with the far environment within its lifetime for very dry environments with low RH values.

As described in the text, the *spatiotemporally evolving* thermal and vapor density gradients in the surrounding air enveloping the droplet play an important role in evolution of the evaporating droplet temperature and radius. Section 8.2 and Fig. 8.4 in Lamb and Verlinde [2011] discuss and show the *steady-state* temperature and vapor profiles around the droplet surface. Lamb and Verlinde [2011] states "The greater the rate of condensation, the greater the temperature increase." (Page 328). As we can see from our simulations in this study, the greater the rate of evaporation for lower $RH_\infty$ values, the greater the difference between the droplet temperature and the far environment. The novelty of our work lies in simulating the evolution of the evolving thermal and vapor density gradients surrounding the evaporating droplet that was never explicitly simulated or considered in these texts or other works.

Egerer, U., Ehrlich, A., Gottschalk, M., Griesche, H., Neggers, R. A. J., Siebert, H., and Wendisch, M.: Case study of a humidity layer above Arctic stratocumulus and potential turbulent coupling with the cloud top, Atmos. Chem. Phys., 21, 6347–6364, https://doi.org/10.5194/acp-21-6347-2021, 2021.

Xue, L., and Coauthors, 2022: Comparison between Observed and Simulated AgI Seeding Impacts in a Well-Observed Case from the SNOWIE Field Program. J. Appl. Meteor. Climatol., 61, 345–367, https://doi.org/10.1175/JAMC-D-21-0103.1.

3.  In figure 4.2, it seems that the droplet's temperature never reaches a steady state. However, since the environment has fixed conditions, it should eventually reach a steady state, provided that the droplet does not evaporate before. How long does it take to reach this steady state in the cases where the droplet has not evaporated yet by the end of the simulation? Somehow it looks like the thermal energy in the droplet is decoupled from the environment and cannot be warmed.

    In our response to the Comment 2, we provided results from two additional simulations that consider droplets evaporating in a high relative humidity environment where the droplets essentially reach steady state before evaporating (the droplet temperature decreases only by one-one hundredth of a degree in 1000s after reaching the environmental wet bulb temperature). The $RH_\infty = 99\%$ condition we simulated for this response is analogous to the condensation calculations in textbooks at supersaturations of 1%. As we noted in the previous response, droplets do not reach steady state for low RH conditions before evaporating.

4.  I was confused by the set-up of the problem. It seems that the authors use cylindrical coordinates instead of spherical coordinates. At the beginning of 3.1., I understood that still the droplet is spherical and it was only the overall domain that was cylindrical, which is fine. But then section 3.4 indicates that boundary conditions are applied at the center of the domain $r = 0$ and not at the center of the droplet, which would mean a whole diameter across the droplet. That was confusing because it seems to indicate that the authors do not consider a spherical droplet but a cylindrical droplet. Is this so? If so, what are the implications in the results? Maybe the authors can clarify this in the paper.

    We have expanded the description of the coordinate systems that are used in COMSOL. For 3D problems, COMSOL has an option to use cartesian, spherical or cylindrical coordinates. For a 2D

axisymmetric domain used in this study, the default coordinate system used by COMSOL is cylindrical coordinates, which in 2D is the same as a cartesian coordinate system in r and z. To avoid any further confusion, we have removed the word cylindrical and used cartesian coordinates.

The physics and the governing equations are the same irrespective of coordinate systems, with the only difference is that all dependent variables (temperature, vapor concentration, etc) will be expressed as functions of r and z. The form of differential operators such as gradient and divergence also changes accordingly. The built-in interface takes care of these changes, along with applying the boundary conditions at appropriate boundaries as indicated. More details regarding the coordinate transformations can be found here. In the end, the results will be independent of coordinate system chosen.

It is actually more complicated since COMSOL uses three coordinate system simultaneously (referred to as the spatial, material and mesh frames). We have added additional information in the text to help the reader understand the Arbitrary Lagrangian-Eulerian framework used in the study. The new text is below:

The simulation of the spatiotemporally varying droplet temperature and radius of an evaporating cloud droplet embedded in a gaseous domain is difficult to solve analytically because of the moving and shrinking boundary at the surface of the evaporating droplet. These kinds of moving boundary problems are also known as Stefan problems. To model this process, we have used an advanced numerical solver, COMSOL (Version 6.0), which employs a finite element method to solve partial differential equations (PDEs). The COMSOL Multiphysics software simultaneously uses spatial, material, and mesh coordinate systems described as the spatial frame, material frame, and mesh frame, respectively. The spatial frame is a fixed, global, Euclidean coordinate system, which in 2D has spatial cartesian coordinates (r, z) with the center of the droplet at (r, z) = (0,0) (Fig. 1). The material frame specifies the material substance, in this case, water or air. The mesh frame is a coordinate system used internally by the finite element method.

The Navier-Stokes and Fick's second law of diffusion equation, which follows from the continuity equation, along with appropriate boundary conditions (see Sec. 3) are solved to conserve mass and momentum in the whole system. The following physics interfaces in COMSOL were used to simulate droplet evaporation: (1) *Two-Phase Laminar Fluid Flow*, which includes a moving mesh to track the shrinking water–air interface of the evaporating water droplet and fluid–fluid interface that incorporates evaporative mass flux; (2) *Transport of Diluted Species* to track water vapor diffusion through the air domain and predict the evaporation rate at the droplet surface; and (3) *Heat Transfer in Fluids* which accounts for the non-isothermal flow within the computational domain, temperature-dependent saturation vapor density at the droplet interface, and a boundary heat source to account for the latent heat of evaporation. The computational domain also includes an infinite element air domain (COMSOL 2023b) to specify and maintain boundary conditions far away from the droplet. The physics modules are coupled through non-isothermal flow between heat transfer and fluid flow, and mass transport at the fluid–fluid interface between fluid flow and species transport.

A non-uniform moving mesh was created by breaking down the computational domain into numerous fine elements of variable sizes, using the Arbitrary Lagrangian-Eulerian technique (Yang et al., 2014) to accurately track the moving air-water interface at the droplet surface. In the ALE technique, the spatial cartesian coordinate system (r, z) is fixed, while the coordinates of the material (R, Z) and the mesh (Rm, Zm) nodes are functions of time as the droplet evaporates. However, the material and mesh node coordinates are always fixed in their respective frames. Initially, the spatial, material and mesh frames are all identical. As the simulation starts, the material and mesh frames deform as the moving boundary of the droplet shrinks during evaporation. After each time step, the

deformed nodes are mapped to the spatial frame, where calculations are performed. In this study, we have used triangular mesh elements (COMSOL 2023c) within the droplet and quadrilateral mesh elements (COMSOL 2023d) for the rest of the domain as shown in Fig. 1. The triangular mesh allows a higher resolution at the droplet surface, and both meshes adjust continually as the droplet surface shrinks during evaporation. Finally, to simulate the water droplet evaporating in ambient air system, with appropriate initial and boundary conditions, the discretized PDEs are numerically solved with adaptive time steps ($\leq 0.01$ s) to maintain numerical stability to obtain the solution (the temporal evolution of droplet temperature and radius) for a range of conditions.

Our wording concerning boundary conditions in the original manuscript was confusing. We replaced the first point with the following:

Within the droplet and throughout the domain, the following conditions are applicable:

$$\boldsymbol{u} \cdot \boldsymbol{n} = 0$$

$$[-\mathrm{p}\boldsymbol{I} + \tau] \cdot \boldsymbol{n} = 0$$

$$\boldsymbol{q} \cdot \boldsymbol{n} = -k\nabla T \cdot \boldsymbol{n} = 0$$

$$-D\nabla c \cdot \boldsymbol{n} = 0$$

where $\boldsymbol{n}$ is the normal to an outward-pointing vector from the center of the droplet. This constraint limits water mass, water vapor and heat flow to the direction normal to the droplet surface.

5. In section 3.2, the equations do not have a number, which makes it difficult to refer to them.

Thank you for pointing this out. We have now added numbers for all equations.

6. In section 3.2, the mass diffusivity retains the dependence on $T$ but the other molecular transport coefficients drop that dependence. Why? If the dependence of $D$ on $T$ plays a role, at least the dependence of the thermal conductivity $k$ on $T$ should be explored, since both transport phenomena are equally important for the problem. Maybe the effect of $T$ on $k$ is negligible compared to that of $T$ on $k$, but then the authors could explain it.

The value of thermal conductivity of water used in the simulations is constant at 0.56 W/(m K). We chose this value based on Fig. 3 of Biddle et al., (2013) where the thermal conductivity of supercooled water is very close to 0.56 W/(m K) for the range of temperatures used in these simulations.

Based on Beard and Pruppacher (1971), the thermal conductivity of air has very weak dependence on temperature over the temperature range used in these simulations. Their thermal conductivity ($k_a$) equation is given by

$$k_a = 0.004184[5.69 + 0.017(T - 273.15)] \quad (\text{W m}^{-1} \text{ K}^{-1})$$

For both T = 273.15 K and 253.15 K, the value of $k_a$ is 0.02 W m$^{-1}$ K$^{-1}$. Hence, we have used a constant value of 0.02 W m$^{-1}$ K$^{-1}$. We have added this information to the paper in Section 3.2:

(2) Heat Transport: The *Heat Transfer in Fluids* interface models heat transfer in all domains (air, water, infinite element domain) using the following version of the heat equation:

$$\rho C_p \frac{\partial T}{\partial t} + \rho C_p \boldsymbol{u} \cdot \nabla T + \nabla \cdot \boldsymbol{q} = 0 \quad (4)$$

$$\boldsymbol{q} = -k \nabla T \qquad (5)$$

where $\rho$ (kg/m³) is the fluid density, $C_p$ (J/(kg·K)) is the fluid heat capacity at constant pressure, $T$ is the temperature, $k$ (W/(m·K)) is the fluid thermal conductivity, $\boldsymbol{u}$ (m/s) is the fluid velocity field from the Laminar Flow interface, $\boldsymbol{q}$ (W/m²) is the heat flux by conduction. We chose the value of $k$ for supercooled water at 0.56 W m⁻¹ K⁻¹ based on Fig. 3 of Biddle et al., (2013) where the thermal conductivity of supercooled water is very close to 0.56 W/(m K) for the range of temperatures used in this study. Based on Beard and Pruppacher (1971), the thermal conductivity of air, given by $k_a = 0.004184[5.69 + 0.017(T - 273.15)]$ (W m⁻¹ K⁻¹), has very weak dependence on temperature over the temperature range used in this study. For both T = 273.15 K and 253.15 K, the value of $k_a$ is 0.02 W m⁻¹ K⁻¹. Hence, we have used a constant value of 0.02 W m⁻¹ K⁻¹.

7. In section 3.2, what is the mathematical expression to calculate $Q_b$?

   The algorithm in COMSOL includes an external heat source or sink term for the air and water domain. We are assuming no external heat source, so the value of $Q_b = 0$. To avoid confusion, we set $Q_b = 0$ so that it no longer appears in the paper.

8. In section 3.2, a reference for the interfacial conditions for the stresses would be useful.

   We have added the reference of Yang et al., (2014) in the paper.

9. In line 392, the authors use the term "the inflection point in the curves," but this might be misleading because those are not inflection points in the usual sense of calculus and curves, i.e., change in the sign of the curvature. Maybe a different term is convenient.

   We have changed the term "the inflection point in the curves" to "the slope transition point" and continue to denote it as $T_i$.

10. In section 4, most of the text repeats what one can see in the figures without adding more insight about the reasons behind that behavior. It might be more helpful to discuss more the physics behind the results.

   We have reduced the number of figures by eliminating the three figures showing the results at $P = 850$ hPa since the results are summarized in Tables 1 and 2. We have expanded our discussion of the remaining figures and the tables in Section 4 to focus on the physics behind the results. In section 4.1, we have added the following paragraph, describing the physics behind the simulations.

   The evaporation process in these experiments starts in a condition that is far from equilibrium. The coupled air-droplet system attempts to evolve towards a steady-state where the thermal energy towards the droplet compensates for evaporative cooling at the droplet surface. In this process, the droplet initially rapidly cools to the thermodynamic wet-bulb temperature of the initial environment similar to what has been shown in Roy et al., (2023). However, under low relative humidity conditions, the thermal and vapor diffusion are not yet near equilibrium. As the system attempts to achieve a steady-state, the imbalance in the heat fluxes associated with vapor and thermal diffusion in the immediate vicinity of the drop leads to a gradual reduction in the wet-bulb temperature of the immediate droplet environment leading to a continued slow decrease in the droplet temperature as the droplet continues to evaporate.

We have also added Section 5.1 in response to review questions #2 and #11.

11. For instance, in section 4.2, the dependence on $T_\infty$ and $P$ seems to be much smaller than that on $RH_\infty$. Why? It might be more useful to differentiate those dependences and concentrate on the latter one instead of showing all figures. Otherwise, there are so many figures that it becomes difficult to distilled the new information that they convey.

    Please see our response to comment #2. As noted above, we eliminated three of the six figures showing results and focused the text more on the physics behind the results.

12. In line 431, the authors write "while the decrease is 5 $K$ in 120 $s$." However, figure 6 seems to indicate 2 $K$ instead of 5 $K$. Is the reference to the figure correct?

    Yes, it's correct. The correct panel to look at is Fig. 5 (d) in the revised manuscript (Fig. 6(d) in the original manuscript), which shows the full 120s evolution. The other two figures (5e, 5f) only show the evolution of temperature at < 1 s. The exact value of decrease in droplet temperature (4.69 K) can be read off Table 1.

13. In that same section, section 4.3, it seems that the main result is that the droplet's temperature is well approximate by the wet-bulb temperature. On the other hand, would not that be expected because of the definition of wet-bulb temperature?

    Also, it is difficult to see something in Fig 8. Why not plot the wet-bulb temperature in figure 4 and 5 and see how it approaches the droplet's temperature at a time $T_i$?

    Rather than clutter figures 4 and 5 (now Figs. 3 and 4) with more lines, we have added Figure 11 to the revised paper showing this evolution for one pressure (500 hPa), temperature (268.15 K), three relative humidities, and three droplet sizes. Just to be clear, the main result is that the droplet's temperature decreases to the wet-bulb temperature directly adjacent to the drop surface which is lower than the wet-bulb temperature of the far environment because of the non-equilibrium of the thermal and vapor fields during the evaporation process. Our previous paper (Roy et al., 2023) showed that at steady-state, the drop temperature can be well-approximated by the thermodynamic wet-bulb temperature of the far environment.

14. Line 489 indicates the "The droplet lifetimes vary widely...", but I am not sure what is meant by widely because the lifetime changes by a factor of 4 while the RH has been changed by a factor of 7. It seems that the lifetime simply follows the change in the control parameter RH.

    We removed the word "widely".

15. Line 496 says "From Figs. 9-14, the decrease in droplet temperatures is independent of the initial droplet size if all other initial environmental conditions are kept constant.". I am not sure if I understand this because fig 9 shows a difference between panels a, d, an g.

    Sorry for the confusion. We have revised the statement to read as follows:

    From Figs. 8-10, for a given initial environmental condition ($RH_\infty$ and $T_\infty$), the droplet temperatures at the end of their lifetimes are independent of the initial droplet sizes.

16. In fig 4 and 5, the initial decrease of $T$ until $T_i$ seems independent of $r_0$ but later on there seems to be a dependence...

Fig. 3 and 4 only show the first 10s of the droplet lifetimes. The differences that the reviewer points out exist because the total lifetimes don't appear on the figure. Table 2 lists the values of the total lifetimes.

17. In line 512, "The decrease in droplet temperatures and increase in droplet lifetimes depict similar relationships with $RH_\infty$ and $r_0$." Why is the dependence on $T_\infty$ smaller than the dependence on other parameters?

    Please see response to Comment #2.

18. Line 634 indicates "the classical pure-diffusion-limited evaporation approach, which ignores evaporative cooling at the droplet surface". I am not sure what is meant because this effect is discussed textbooks like Rogers and Yau [1989] and Lamb and Verlinde [2011] which are often used to teach microphysics.

    By "classical", we mean Maxwellian approach (Maxwell, 1890; Eq 13-10 of Pruppacher and Klett, 1997), which is what is discussed in relationship to condensation in both Rogers and Yau [1989] and Lamb and Verlinde [2011]. In that approach, the droplet temperature is assumed to be in equilibrium with the ambient environment. Again, these textbooks don't address the evaporation problem with regards to evolving cloud droplet temperature and lifetime. We changed the term "classical" to Maxwellian to be clear what we are referring to.

19. Line 669 seems to contain the main message "Comparing these values with reported droplet freezing timescales available from experimental studies, droplet freezing events can potentially occur within the time frame when these droplets can reach lower temperatures due to evaporative cooling before they completely dissipate into the subsaturated air." However, this message was a bit lost in the middle of a long paragraph. Maybe it helps to make the statement at the beginning and then elaborate it.

    Thank you for your suggestion. We have reorganized the paragraph to move this point toward the front.

20. In the conclusions, the first sentence says "...internal thermal gradients within the droplet as well as resolving thermal and vapor density gradients in the surrounding ambient air." suggesting that this is new. However, these gradients are retained in the classical studies, e.g., Rogers and Yau [1989] and Lamb and Verlinde [2011]. Do the authors mean the temporal variation of those gradients?

    The reviewer is correct. We have changed the sentence to read:

    In this study, we presented a quantitative investigation of the temperature and lifetime of an evaporating droplet, considering internal thermal gradients within the droplet as well as resolving spatiotemporally varying thermal and vapor density gradients in the surrounding ambient air.

**References**

D. Lamb and J. Verlinde. *Physics and Chemistry of Clouds*. Cambridge University Press, 2011.

R. R. Rogers and M. K. Yau. *A Short Course in Cloud Physics*. Butterworth-Heinemann, third edition, 1989.

---

## Author Comment (AC3)

**Response to Reviewer 2**

We thank the reviewer for providing insightful comments and suggestions on our paper. We believe that responding to the comments has further improved the quality of the manuscript. We have addressed the comments below. The reviewer comments are in **black**, our responses in **red** and any text changes in **blue**.

**Review 2**

This work quantitatively investigated the evolution of an evaporating droplet and its surrounding environment utilizing an idealized numerical model. With different settings of ambient temperature, relative humidity, pressure, and initial droplet radius, this work examines the evolution of droplet temperature and lifetime and their dependences with these environment factors. The findings confirmed the previous literature that assumes steady-state droplet temperature, and the main novelty is finding that droplets can be much colder and last longer, due to the cooling of the adjacent air and the temperature gradient in the immediate environment surrounding the droplets. The results of this study is particularly of interest to the modeling community, which has been struggling with the underestimation of INPs for a long time. I think this manuscript is generally well written and recommend its publication in ACP, with some comments listed below.

Detailed comments:

1.  Line 31: what do you mean by "cells"? do you mean a grid in numerical models?

    By "ice-generating cells", we refer to small convective cells at tops of otherwise stratiform clouds. The American Meteorological Society's Glossary of Meteorology defines "generating cells" as "In radar, a small region of locally high reflectivity from which a trail of hydrometeors originates". The term was first used in 1953 by J.S. Marshall (Marshall, J. S., 1953: Precipitation trajectories and patterns. J. Atmos. Sci., 10, 25–29, doi:10.1175/1520-0469(1953)010<0025:PTAP>2.0.CO;2.). A good review of generating cells can also be found in Plummer et al., 2014, included in our reference list.

    To avoid any further ambiguity, we have now added a reference to the AMS Glossary definition in the revised text.

2.  Line 70-72: this sentence has too many sub-sentences, suggesting rewrite it.

    We have simplified the text to read as follows:

    Lü et al., (2005) conducted ice nucleation experiments with acoustically levitated supercooled water droplets. Using statistical analyses of nucleation rates, they found that ice nucleation predominantly initiates in the vicinity of the droplet surface.

3.  Line 151: I am curious if there are any differences between using cylindrical coordinates and spherical coordinates in the model, as the droplet volume, surface curvature and water tension may be calculated differently in the two coordinates.

    We repeat our response to Reviewer 1 who also had similar concerns.

    We have expanded the description of the coordinate systems that are used in COMSOL. For 3D problems, COMSOL has an option to use cartesian, spherical or cylindrical coordinates. For a 2D

axisymmetric domain used in this study, the default spatial coordinate system used by COMSOL is cylindrical coordinates, which in 2D is the same as a cartesian coordinate system in r and z. To avoid any further confusion, we have removed the word cylindrical and used cartesian coordinates instead.

The physics and the governing equations are the same irrespective of coordinate systems, with the only difference is that all dependent variables (temperature, vapor concentration, etc) will be expressed as functions of r and z. The form of differential operators such as gradient and divergence also changes accordingly. The built-in interface takes care of these changes, along with applying the boundary conditions at appropriate boundaries as indicated. More details regarding the coordinate transformations can be found here. In the end, the results will be independent of coordinate system chosen.

It is actually more complicated since COMSOL uses three coordinate systems simultaneously (referred to as spatial, material and mesh frames). We have added additional information in the text (Section 2.1) to help the reader understand the Arbitrary Lagrangian-Eulerian (ALE) framework utilized in this study to track the moving boundary of the shrinking droplet. The revised text is also given below:

The simulation of the spatiotemporally varying droplet temperature and radius of an evaporating cloud droplet embedded in a gaseous domain is difficult to solve analytically because of the moving and shrinking boundary at the surface of the evaporating droplet. These kinds of moving boundary problems are also known as Stefan problems. To model this process, we have used an advanced numerical solver, COMSOL (Version 6.0), which employs a finite element method to solve partial differential equations (PDEs). The COMSOL Multiphysics software simultaneously uses spatial, material, and mesh coordinate systems described as the spatial frame, material frame, and mesh frame, respectively. The spatial frame is a fixed, global, Euclidean coordinate system, which in 2D has spatial cartesian coordinates (r, z) with the center of the droplet at (r, z) = (0,0) (Fig. 1). The material frame specifies the material substance, in this case, water or air. The mesh frame is a coordinate system used internally by the finite element method.

The Navier-Stokes and Fick's second law of diffusion equation, which follows from the continuity equation, along with appropriate boundary conditions (see Sec. 3) are solved to conserve mass and momentum in the whole system. The following physics interfaces in COMSOL were used to simulate droplet evaporation: (1) *Two-Phase Laminar Fluid Flow*, which includes a moving mesh to track the shrinking water–air interface of the evaporating water droplet and fluid–fluid interface that incorporates evaporative mass flux; (2) *Transport of Diluted Species* to track water vapor diffusion through the air domain and predict the evaporation rate at the droplet surface; and (3) *Heat Transfer in Fluids* which accounts for the non-isothermal flow within the computational domain, temperature-dependent saturation vapor density at the droplet interface, and a boundary heat source to account for the latent heat of evaporation. The computational domain also includes an infinite element air domain (COMSOL 2023b) to specify and maintain boundary conditions far away from the droplet. The physics modules are coupled through non-isothermal flow between heat transfer and fluid flow, and mass transport at the fluid–fluid interface between fluid flow and species transport.

A non-uniform moving mesh was created by breaking down the computational domain into numerous fine elements of variable sizes, using the Arbitrary Lagrangian-Eulerian technique (Yang et al., 2014) to accurately track the moving air-water interface at the droplet surface. In the ALE technique, the spatial cartesian coordinate system (r, z) is fixed, while the coordinates of the material (R, Z) and the mesh ($R_m$, $Z_m$) nodes are functions of time as the droplet evaporates. However, the material and mesh node coordinates are always fixed in their respective frames. Initially, the spatial, material and mesh frames are all identical. As the simulation starts, the material and mesh frames

deform as the moving boundary of the droplet shrinks during evaporation. After each time step, the deformed nodes are mapped to the spatial frame, where calculations are performed. In this study, we have used triangular mesh elements (COMSOL 2023c) within the droplet and quadrilateral mesh elements (COMSOL 2023d) for the rest of the domain as shown in Fig. 1. The triangular mesh allows a higher resolution at the droplet surface, and both meshes adjust continually as the droplet surface shrinks during evaporation. Finally, to simulate the water droplet evaporating in ambient air system, with appropriate initial and boundary conditions, the discretized PDEs are numerically solved with adaptive time steps ($\leq$ 0.01 s) to maintain numerical stability and obtain the solution (the temporal evolution of droplet temperature and radius) for a range of conditions.

4. Line 170-171: Can you add a few sentences describing why it uses different meshes in and out of the droplet? any pros and cons for this setting?

The triangular mesh allows a higher resolution at the droplet surface and the mesh adjusts continually as the droplet surface shrinks during evaporation. We have now added this information to the revised paper.

5. Line 190-191: maybe change the temperature unit from K to C for easier read. Same as the figures and tables.

We have considered this option but decided to keep all the temperatures in Kelvin for ease of comparison with our previous paper (Roy et al., 2023). In addition, regeneration of all the figures and tables and text would be difficult in the given timeframe to submit the revised paper.

6. Line 211-213: related to my comment #3, maybe this is something can be used to explain that using cylindrical coordinates is appropriate.

Please see our response to Comment #3.

7. Line 252: give a number to the equations.

All equations are now numbered.

8. Line 257: μ should have a value, what is the number?

COMSOL uses temperature dependent empirical formulae for the dynamic viscosity of water and air. However, for water below 273.15 K, the dynamic viscosity is approximated as 1.79 mPa s. For air, the equation is an empirical equation that produces values equivalent to Sutherland's law.

We have added this information to the revised paper.

"….and $\mu$ is the fluid dynamic viscosity. For water below 273.15 K, the dynamic viscosity can be approximated as 1.79 mPa s. For air, COMSOL uses an empirical equation that produces values equivalent to Sutherland's law (White, 2006), $\mu = \mu_0 \left(\frac{T}{T_0}\right)^{\frac{3}{2}} \left(\frac{T_0 + S_\mu}{T + S_\mu}\right)$

where $\mu_0$ = 1.716 × 10⁻⁵ N s m⁻², $T_0$ = 273 K, and $S_\mu$ = 111 K for air. The empirical equation is given as:

$$\mu = -8.38278 \times 10^{-7} + 8.35717342 \times 10^{-8}T - 7.69429583 \times 10^{-11}T^2 + 4.6437266 \times 10^{-14}T^3 - 1.06585607 \times 10^{-17}T^4 \qquad (4)"$$

9. Line 268: k should be a constant or function depending on T and p, what is the number? And is it different in the droplet and in the environment air?

The value of thermal conductivity of water used in the simulations is constant at 0.56 W/(m K). We chose this value based on Fig. 3 of Biddle et al, (2013) where the thermal conductivity of supercooled water is very close to 0.56 W/(m K) for the range of temperatures in the simulations. We have added information and a reference to Biddle et al., (2013) in the revised paper.

Based on Beard and Pruppacher (1971), the thermal conductivity of air has very weak dependence on temperature over the temperature range used in the simulations. Their thermal conductivity ($k_a$) equation is given by

$$k_a = 0.004184[5.69 + 0.017(T - 273.15)] \quad (\text{W m}^{-1} \text{ K}^{-1})$$

For both T = 273.15 K and 253.15 K, the value of $k_a$ is 0.02 W m$^{-1}$ K$^{-1}$. Hence, we have used a constant value of 0.02 W m$^{-1}$ K$^{-1}$. We have added this information to the revised paper:

We chose the value of $k$ for supercooled water at 0.56 W m$^{-1}$ K$^{-1}$ based on Fig. 3 of Biddle et al., (2013) where the thermal conductivity of supercooled water is very close to 0.56 W/(m K) for the range of temperatures used in this study. Based on Beard and Pruppacher (1971), the thermal conductivity of air, given by $k_a = 0.004184[5.69 + 0.017(T - 273.15)]$ (W m$^{-1}$ K$^{-1}$), has very weak dependence on temperature over the temperature range used in this study. For both T = 273.15 K and 253.15 K, the value of $k_a$ is 0.02 W m$^{-1}$ K$^{-1}$. Hence, we have used a constant value of 0.02 W m$^{-1}$ K$^{-1}$.

10. Line 276: one factor that impacts the final temperature drop at the droplet surface is the difference of water diffusivity and heat diffusivity of the environment air. I am wondering how large the diffusivity uncertainties of water and heat are, and how this will impact the temperature drop.

We tried to find information on uncertainties in these values. Biddle et al, (2013) doesn't discuss uncertainties. Unfortunately, we were unable to find information the reviewer requests. We can assume that that the uncertainty is no larger than the last significant figure reported for these values, which suggests very high accuracy given the number of significant figures reported.

11. Line 292: $T_\infty$ should be in the unit of K when multiplied with R.

The reviewer is correct and that was what was done. However, the way the sentence was phrased was misleading. The sentence originally read:

For the vapor transfer interface, except within the droplet, all domains are at an initial vapor concentration of $c_{0,air}$ which is again assumed to be the same as that of the constant ambient concentration value far from the droplet, $c_\infty$, calculated as follows:

$c_\infty = \dfrac{RH_\infty \times e_{S T_\infty}}{R_{univ} \times T_\infty}$ where, $RH_\infty$ is set at a constant ambient relative humidity far from the droplet, $R_{univ}$ = 8.3145 (J/mol/K) and saturation vapor pressure, $e_{S T_\infty} = 610.94 * \exp\left(\dfrac{17.625 * T_\infty}{T_\infty + 243.04}\right)$ (in Pa, with $T_\infty$ in °C) following Alduchov and Eskridge (1996).

We have now changed the sentence in the revised paper to read:

For the vapor transfer interface, except within the droplet, all domains are at an initial vapor concentration of $c_{0,air}$ which is again assumed to be the same as that of the constant ambient concentration value far from the droplet, $c_\infty$, calculated as follows:

$c_\infty = \frac{RH_\infty \times e_{S T_\infty}}{R_{univ} \times T_\infty}$ where, $RH_\infty$ is set at a constant ambient relative humidity far from the droplet, $R_{univ}$ = 8.3145 (J/mol/K), $T_\infty$ is in K. The saturation vapor pressure is calculated as, $e_{S T_\infty}$ = 610.94 * $\exp\left(\frac{17.625 * T_\infty}{T_\infty + 243.04}\right)$ (in Pa, with $T_\infty$ in °C) following Alduchov and Eskridge (1996).

12. Line 309: again, T should be in the unit of K here.

The reviewer is correct and that was what was done. The way the sentence was phrased was misleading. The sentence was changed in the revised paper to read as follows:

Hence, saturated vapor concentration at the shrinking droplet boundary, using the ideal gas law, is given by, $c_{sat}(T_{sf}) = \frac{e_s(T_{sf})}{R_{univ} \times T_{sf}}$ where $T_{sf}$ is the surface temperature, in K. The saturation vapor pressure $e_s(T_{sf})$ is estimated as $e_s(T_{sf})$ = 610.94 * $\exp\left(\frac{17.625 * T_{sf}}{T_{sf} + 243.04}\right)$ (in Pa, with $T_{sf}$ in °C) again following Alduchov and Eskridge (1996).

13. Line 396: the number of mean cooling rate (K/s) is huge but does not mean anything, it is just an initial model spinup. Maybe just remove it.

Thank you for the suggestion. We have removed all instances of mean cooling rate.

14. Line 399-401: These numbers are different from the numbers in Figure 4.

The numbers in the original Figure 4 (now Fig. 3) for final temperature of the droplets were averages of the three values for $r_0$ = 10, 30 and 50 µm because it was hard to fit all three numbers on the figure. We have changed the caption to make that clear. The caption now reads,

Droplet temperature evolution (left column) and radius evolution (right column) for three different $RH_\infty$ ($RH_\infty$ = 10% (brown curves), 40% (orange curves) and 70% (green curves)), three different $r_0$ ($r_0$ = 10 µm (dot-dashed lines), 30 µm (solid lines) and 50 µm (dashed lines)), with three different $T_\infty$ = 273.15 K (0°C) (a, b), 268.15 K (-5°C) (c, d) and 263.15 K (-10°C) (e, f), for $P$ = 500 hPa. For each $RH_\infty$, the average droplet temperature at the end of the lifetimes of the three droplets with different $r_0$ ($T_L$, in K) is given in (a,c,e) and the time taken to reach the end of its lifetime ($t_L$, in s) is given in (b, d, f). Exact values of final temperature for each $r_0$ are given in Table 1.

15. Figure 4: again, I suggest using C instead of K for the unit of temperature. This makes y axis cleaner.

As discussed earlier, we had considered this option but decided to keep all the temperatures in Kelvin for ease of comparison with our previous paper (Roy et al., 2023). In addition, regeneration of all the figures and tables and text would be difficult in the given timeframe to submit the revised paper.

16. Section 4.4: The presentation in this section needs to be improved. The authors list many numbers for different conditions, easily making readers get lost which quantity is in comparison (e.g., Section 4.4.2). I strongly suggest the authors simplify the text. For example, saying that "For environment with RH=10%, T=273K, P=500hPa, the lifetimes of 10, 30, 50 um diameter droplet are 1.1s, 1.4s,

32.8s, respectively." (well, the effect of droplet size to lifetime is obvious, maybe section 4.4.2 can be removed or modified).

In response to this comment and reviewer 1's concern, we have reduced the number of figures by eliminating the three figures showing the results at $P = 850$ hPa since the results are summarized in Tables 1 and 2. We have expanded our discussion of the remaining figures and the tables in Section 4 to focus on the physics behind the results.

17. Table 1: again, using C instead of K makes it easier to read.

Please see our previous response (Comment 5).

18. Table 2: I would not put lifetime difference ($t_L$ - $t_{LC}$) in the table, or just use a relative difference (percentage change), which is more relevant to the modeling application.

Thank you for the suggestion. We have added two extra columns to Table 2 that show the percentage changes.

19. Line 592-605: This paragraph may need to be re-organized or re-stated. It currently reads like saying the previous assumption of steady-state droplet temperature is imperfect and this study improves it. However, this study verified that the steady-state droplet temperature assumption is valid, with the main novelty to be considering the gradient of adjacent environment, which was not considered in previous studies.

We tried to soften the paragraph, so it didn't give the impression that we are criticizing past work. We have removed the phrase "In order to model a more realistic scenario of an isolated droplet evaporating in a subsaturated environment". We note that our study showed that the steady-state droplet temperature assumption is not valid for conditions where droplets are evaporating in environments with moderate to low relative humidity. In very high relative humidity environments, approximately > 90%, the steady-state assumption can provide a reasonable estimation of the droplet temperature.

To be clear, we showed that the steady-state droplet temperature assumption is not always valid. This can be directly seen by comparing the $T_{RRD}$ column (the steady-state temperature) with $T_L$ column (temperature from the current simulations) in Table 1. For the range of ambient relative humidities we simulated (10-70%), the droplet temperature deviates from the steady-state simulations. For example, for 50 μm droplet, for an ambient temperature of 263.15 K, the temperature deviation from the steady state-solution can vary from 2.6 K for $RH_\infty$ = 70% to 16.2 K for $RH_\infty$ = 10%. Even for the smaller deviation, the impact on potential ice nucleation events can be significant, because of the strong dependence of ice-nucleation rates on temperature. As noted in response to the first reviewer, at very high relative humidities (e.g. $RH_\infty$ = 99%) the steady-state assumption is valid.

20. Line 623: I am curious whether the RH = 10% is realistic in real world. In another words, do we really have a droplet ~ 25 K colder than we thought?

In situations where air has a history of descent above a stratiform cloud deck, the vertical humidity gradient can be quite large. Observations of relative humidity vertical profiles above Arctic stratocumulus clouds reveal strong vertical gradients in RH at cloud top, with RH values as dry as ~ 50% (Egerer et al., 2021). Also, during the SNOWIE field campaign, dry layer incursion (with RH ~10-20%) was observed above orographic clouds, which led to very sharp gradients in RH at cloud

tops (Xue et al. 2022). Here, in this study, we have chosen 10% and 70% to bound the simulations on both ends.

Egerer, U., Ehrlich, A., Gottschalk, M., Griesche, H., Neggers, R. A. J., Siebert, H., and Wendisch, M.: Case study of a humidity layer above Arctic stratocumulus and potential turbulent coupling with the cloud top, Atmos. Chem. Phys., 21, 6347–6364, https://doi.org/10.5194/acp-21-6347-2021, 2021.

Xue, L., and Coauthors, 2022: Comparison between Observed and Simulated AgI Seeding Impacts in a Well-Observed Case from the SNOWIE Field Program. J. Appl. Meteor. Climatol., 61, 345–367, https://doi.org/10.1175/JAMC-D-21-0103.1.

21. Line 633: it also includes 10 μm droplet.

Thank you for pointing this out. We have now added the 10 μm droplet to the sentence in the revised paper.

---

## Author Response (AR2)

**Response to reviews**

We thank the reviewer for their comments which have improved the quality of our manuscript.

**Reviewer 1**

One thing that I still think might be worth sharpening in the introduction and discussion of the paper is the reference to the previous work having not considered evaporation and the effect of the temperature variation.

Response: In the original paper, we referred to the comprehensive review of past theoretical, experimental and numerical studies of droplet evaporation presented in Roy et al., 2023. This was the reason we initially didn't include the fundamental material requested by the reviewer. In this revised version, we have added an additional paragraph to the paper in the Introduction to briefly summarize the key literature requested by the reviewer.

As the authors indicate, textbooks often consider evaporation as it occurs in a falling droplet, but the same equations hold as in the case of condensation, and a similar interpretation of the mechanisms described by those equations (changing of course from supersaturation to subsaturation, condensational warming to evaporative cooling, and so...). The authors refer to Pruppacher and Klett. In Rogers and Yau, it is in the last paragraph of the first section of chapter 7, page 105 in the 3rd edition, which starts "The rate of evaporation of a droplet is also described by (7.18)...". Those equations retain the radial variation and the effect of latent heat (evaporative cooling in case of evaporation) on the droplet's temperature, $T_r$, which is then different from the ambient temperature, $T\_infty$, and retaining this difference in the saturation vapor pressure changes the evaporation rate by a factor of order one.

Response: The last paragraph of Rogers and Yau (1987) is focused on droplet fall velocity as a function of droplet size. Eq. 7.18 is an approximate solution to the droplet growth/evaporation equation first presented in Mason (1971). This quasi-steady approximation is applicable for droplet growth where the supersaturation is typically less than 1% and the difference between the droplet temperature and ambient air is negligible. For evaporation, where vapor deficits can occur over a wide range of relative humidities, the approximation breaks down as the droplet temperature can deviate significantly from the ambient environment (Srivastava and Coen, 1992; Roy et al., 2023).

We have added a statement to this effect in the Introduction.

Sometimes in the text, I think that these ideas are being mixed with the quasi-steady approximation that is assumed in those textbooks, which might obscure the novelty of this paper, namely, the investigation of unsteady effects. That is why think that sharpening the differences to that previous work might be worth to strengthen the paper.

Response: We have added a paragraph addressing the reviewer's concern in the Introduction and noted in the Introduction and the Conclusions that the focus of this study is on the unsteady solution of droplet evaporation in a subsaturated environment.